# Informed Asymmetric Actor-Critic:
# Leveraging Privileged Signals Beyond Full-State Access

**Daniel Ebi** [1]   **Damien Ernst** [2]   **Klemens Böhm** [1]   **Gaspard Lambrechts** [3] [4]

## Abstract

Asymmetric reinforcement learning leverages privileged information available during training to improve learning under partial observability. Existing asymmetric actor-critic methods typically assume access to the full environment state to condition the critic during training, which is often unrealistic in practice. We introduce the informed asymmetric actor-critic framework that allows the critic to be conditioned on arbitrary state-dependent privileged signals, and show that any such signal yields unbiased policy gradient estimates. This substantially expands the set of admissible privileged information and raises the problem of selecting the most informative signals for learning. To this end, we propose two novel informativeness criteria: a dependence-based test that can be applied prior to training, and a test based on improvements in value prediction that can be applied post hoc. Experiments on partially observable benchmarks and synthetic environments demonstrate that carefully selected privileged signals can match or outperform full-state asymmetric baselines while relying on strictly less state information.

## 1. Introduction

Reinforcement learning (RL) has emerged as a powerful tool for optimizing control policies in various domains, including heating, ventilation, and air conditioning systems (al Sayed et al., 2024), energy systems (François-Lavet et al., 2016; Ebi et al., 2024), autonomous driving (Sallab et al., 2017), and robotics (Tang et al., 2025). However, many real-world applications involve partial observability, where agents must make decisions based on incomplete and noisy observations. This setting is formalized by partially observable Markov decision processes (POMDPs) (Kaelbling et al., 1998), where optimal actions depend on the history of past observations and actions.

To address this, RL methods for fully observable settings have been adapted by learning history-dependent policies, typically using recurrent neural networks (RNNs) that encode observation-action sequences (Bakker, 2001; Wierstra et al., 2010; Hausknecht & Stone, 2015; Zhang et al., 2016; Zhu et al., 2017; Cayci & Eryilmaz, 2025). Many approaches compress these sequences into compact latent representations by introducing auxiliary learning objectives (Igl et al., 2018; Buesing et al., 2018; Guo et al., 2018; Gregor et al., 2019; Han et al., 2020; Guo et al., 2020; Lee et al., 2020; Subramanian et al., 2022; Ni et al., 2024). While these methods can, in principle, learn optimal policies, they assume the same level of observability during training and execution, restricting policy learning to the limited information available at deployment. Yet, in practice, this assumption is unnecessarily restrictive and possibly suboptimal. Many training environments provide additional information that is unavailable or impractical to access during execution, including multi-view sensory input in robotics (Pinto et al., 2018), simulator states available during training in simulation, expert policies, and external knowledge obtained from foundation models by querying representations conditioned on observations and eventual additional context. These signals do not necessarily correspond to the full state or satisfy the Markov property, yet they can provide useful information for improving learning.

Several approaches have explored the use of privileged information. For example, some methods train privileged expert policies conditioned on the true state and then imitate them (Choudhury et al., 2018). However, these methods often lack theoretical guarantees and may lead to suboptimal policies in POMDPs (Warrington et al., 2021). To mitigate this, Warrington et al. (2021) propose constraining the expert policy to yield an optimal policy under partial observability. Another line of work exploits privileged information in model-based RL by constructing world models

[1]Department of Computer Science, Karlsruhe Institute of Technology, Karlsruhe, Germany [2]Montefiore Institute, University of Liège, Liège, Belgium [3]Department of Electrical and Computer Engineering, McGill University, Montreal, Canada [4]Mila - Québec Artificial Intelligence Institute, Montreal, Canada. Correspondence to: Daniel Ebi <daniel.ebi@kit.edu>.

*Proceedings of the 43rd International Conference on Machine Learning*, Seoul, South Korea. PMLR 306, 2026. Copyright 2026 by the author(s).

that summarize by integrating additional state signals. Examples include the Informed Dreamer (Lambrechts et al., 2024), the Wasserstein Believer (Avalos et al., 2024), and the Scaffolder (Hu et al., 2024).

Finally, asymmetric actor-critic methods offer a promising approach for leveraging privileged information by conditioning the actor on observable histories and the critic on privileged signals during training (Degrave et al., 2022; Kaufmann et al., 2023; Vasco et al., 2024; Dürr et al., 2026). Performance improvements arise from better value estimation rather than richer actor inputs, as the actor does not access privileged signals. Pinto et al. (2018) introduce an early asymmetric actor-critic method with a state-conditioned critic that achieves strong empirical performance. However, this formulation is generally ill-defined in POMDPs unless additional assumptions hold, such as state-decodability, where the belief over latent states collapses to a Dirac distribution given the history (Baisero & Amato, 2022). Baisero & Amato (2022) address this issue by introducing the history-state value function to ensure unbiased gradients. Theoretical work has further established convergence guarantees for policy gradient and actor-critic methods in both fully and partially observable settings, including symmetric recurrent natural actor-critic methods using RNNs (Wang et al., 2020; Agarwal et al., 2021; Cayci & Eryilmaz, 2025) and asymmetric settings with fixed agent state and linear function approximators (Lambrechts et al., 2025) or tabular belief-weighted formulations (Cai et al., 2024). However, existing asymmetric actor-critic approaches often assume full-state access, leaving the case of privileged partial signals underexplored. A related line of work in causal decision-making and bandit settings studies when optimal policies depend only on subsets of variables (Lee & Bareinboim, 2018; Lu et al., 2020; Lee & Bareinboim, 2020). These approaches typically rely on an explicit causal model to identify irrelevant variables. In contrast, we do not assume access to such a structure and instead study privileged signals from a statistical perspective focused on their utility for value estimation.

In this work, we generalize asymmetric actor-critic methods by introducing the informed asymmetric actor-critic framework, which allows the critic to be conditioned on arbitrary state-dependent privileged signals without requiring full-state access. We show that such state-conditioned signals can be leveraged while preserving unbiased policy gradients. This substantially broadens the class of admissible privileged training-time signals and extends the scope of asymmetric actor-critics, but also raises a natural question: if any state-conditioned signal is admissible, which ones should be selected?

Since there is no inherent criterion for preferring one admissible signal over another, the selection and evaluation of use-ful signals becomes a central challenge in asymmetric RL. We provide a first systematic answer to this question by introducing two informativeness criteria that assess the utility of additional signals w.r.t. value estimation: (i) a residual-based dependency measure, and (ii) a value-prediction-based measure. We empirically evaluate our framework on benchmark environments and synthetic informed POMDPs. Our results show that leveraging informative privileged signals can significantly improve policy learning, challenging the assumption that full-state access is essential for effective asymmetric actor-critics. This work suggests new directions for studying how privileged information can be selected, evaluated, and integrated to facilitate learning in partially observable settings.

## 2. Background

In this section, we formally introduce partially observable Markov decision processes (POMDPs) and the symmetric actor-critic paradigm, and then describe the informed POMDP framework that motivates our informed asymmetric actor-critic framework.

### 2.1. Partially Observable Markov Decision Processes

A POMDP (Kaelbling et al., 1998) models sequential decision-making under uncertainty as tuple $(\mathcal{S}, \mathcal{A}, \mathcal{O}, T, O, R, P, \gamma)$, where $\mathcal{S}$, $\mathcal{A}$, and $\mathcal{O}$ denote the state, action, and observation spaces. The transition probabilities $T(s_{t+1}|s_t, a_t)$ describe the process dynamics. The environment emits observations via $O(o_t|s_t)$ at time $t$ and selects actions $a_t$ based on the observable history $h_t$, defined as the sequence of past observations and actions, including the current observation. Specifically, the history can be written as $h_t = (o_0, a_0, \ldots, o_t)$. We define the set of observable histories as $\mathcal{H} = \bigcup_{t=0}^{\infty} \mathcal{H}_t$, where $\mathcal{H}_t \subseteq \mathcal{O} \times (\mathcal{A} \times \mathcal{O})^t$ is the set of histories of size $t$. The immediate reward of the agent is given by $r_t := R(s_t, a_t)$, with $R: \mathcal{S} \times \mathcal{A} \to \mathbb{R}$. We assume that there exists a constant $r_{\max} > 0$ such that $|r_t| \leq r_{\max}$ for all $t$. $P$ specifies the initial state distribution. The objective is to maximize the expected return $J(\pi) = \mathbb{E}^{\pi}\left[\sum_{t=0}^{\infty} \gamma^t R(s_t, a_t)\right]$, where $\pi(a_t|h_t)$ denotes a history-dependent policy and $\gamma \in [0, 1)$ is a discount factor. The future return from time $t$ is defined as $G_t = \sum_{k=0}^{\infty} \gamma^k R(s_{t+k}, a_{t+k})$, which leads to the history-based $Q$-function, the expected return given the current history and action,

$$Q^{\pi}(h_t, a_t) = \mathbb{E}_{s_{t:\infty}, a_{t+1:\infty}}^{\pi}\left[G_t|h_t, a_t\right],$$

and the corresponding history value function

$$V^{\pi}(h_t) = \sum_{a_t \in \mathcal{A}} \pi(a_t|h_t) \, Q^{\pi}(h_t, a_t).$$

## 2.2. Symmetric Actor-Critic Paradigm

Given a history-dependent policy $\pi_\theta$ parametrized by $\theta$ (actor), the policy gradient in a POMDP is given by (Sutton et al., 1999; Wierstra et al., 2007)

$$\nabla_\theta J(\pi_\theta) = \mathbb{E}\left[\sum_t \gamma^t Q^\pi(h_t, a_t) \nabla_\theta \log \pi_\theta(a_t|h_t)\right]. \quad (1)$$

In practice, this expectation is estimated via Monte Carlo by sampling histories and actions in the POMDP, which is known to yield high-variance estimates. It has been shown that subtracting any function of the history, referred to as a baseline, from the $Q$-function does not bias the gradient while potentially reducing variance. Commonly, the value function $V^\pi(h_t)$ is used as this baseline, yielding the advantage formulation

$$\nabla_\theta J(\pi_\theta) = \mathbb{E}\left[\sum_t \gamma^t A^\pi(h_t, a_t) \nabla_\theta \log \pi_\theta(a_t|h_t)\right], \quad (2)$$

where $A^\pi(h_t, a_t) = Q^\pi(h_t, a_t) - V^\pi(h_t)$ is called the advantage function. Unfortunately, the exact forms in Equations 1-2 involve the unknown functions $Q^\pi(\cdot)$ and $V^\pi(\cdot)$. The $Q$-function $Q^\pi(h_t, a_t)$ can in principle be estimated using a single Monte Carlo sample of $G_t$, but $V^\pi(h_t)$ cannot because the sampled $G_t$ is not conditionally independent of $a_t$ given $h_t$. Practitioners typically approximate $Q^\pi(\cdot)$ or $V^\pi(\cdot)$ with parametric functions $Q_\vartheta^\pi$ or $V_\vartheta^\pi$ (critic), using temporal difference (TD) learning. Since a baseline requires $V^\pi(\cdot)$, many algorithms avoid estimating $Q^\pi$ directly and instead rely on an advantage estimator that uses only the critic, $\hat{A}_t = r_t + \gamma V_\vartheta^\pi(h_{t+1}) - V_\vartheta^\pi(h_t)$, where $r_t$ is the observed reward at time $t$. This estimator of the advantage is unbiased if $V_\vartheta^\pi$ perfectly approximates $V^\pi$.

## 2.3. Informed POMDPs

Modeling the training-execution asymmetry, in which the critic accesses additional state-dependent signals during training, but the policy relies only on observation-action histories, requires a framework that formalizes these signals without altering execution-time dynamics. The informed POMDP formalism (Lambrechts et al., 2024) provides a principled approach by augmenting the standard POMDP with an information variable. Specifically, the informed POMDP introduces an information space $\mathcal{I}$ and an information function $I\colon \mathcal{S} \to \Delta(\mathcal{I})$, which gives the probability to obtain information $i_t \in \mathcal{I}$ in the true state $s_t \in \mathcal{S}$. Hence, the informed POMDP is defined by the 10-tuple $(\mathcal{S}, \mathcal{A}, \mathcal{I}, \mathcal{O}, T, I, \widetilde{O}, R, P, \gamma)$. In contrast to the standard POMDP, the observation function is defined as $\widetilde{O}\colon \mathcal{I} \to \Delta(\mathcal{O})$ and denotes the probability to obtain $o_t \in \mathcal{O}$ given $i_t \in \mathcal{I}$. The defining assumption of the informed POMDP is that the observation $o_t$ is conditionally independent of the true state $s_t$ given the infor-

mation $i_t$, i.e., $p(o_t|i_t, s_t) = \widetilde{O}(o_t|i_t)$.[1] Each informed POMDP induces an underlying execution POMDP defined as $(\mathcal{S}, \mathcal{A}, \mathcal{O}, T, O, R, P, \gamma)$, with observation function $O(o_t|s_t) = \mathbb{E}_{i_t|s_t}[\widetilde{O}(o_t|i_t)]$, which governs the agent's interaction at deployment.

Thus, the conditional-independence assumption offers a convenient modeling abstraction for introducing an information variable that is at least as informative about the underlying state as the execution-time observation, while preserving full generality. As a result, the informed POMDP framework naturally subsumes both standard POMDPs and settings with additional training-time signals, making it particularly well-suited for modeling asymmetric learning scenarios.

# 3. Informed Asymmetric Actor-Critic

In this section, we develop an asymmetric actor-critic framework that leverages arbitrary state-conditioned information during training. Building on the informed POMDP paradigm, we introduce informed history-based $Q$- and value functions and derive an unbiased policy gradient estimator. Our formulation generalizes the history-state critic of Baisero & Amato (2022): instead of requiring full-state access, we show that any privileged signal $i_t \sim I(i_t|s_t)$ suffices to obtain unbiased value estimates, without imposing additional assumptions on the POMDP. This expands the set of usable training-time signals in asymmetric actor-critics, motivating the question of how to select among them.

## 3.1. Informed History-based Value Functions

Given an informed POMDP, we define the following informed history-based reward

$$R(h_t, i_t, a_t) = \mathbb{E}_{s_t}\left[R(s_t, a_t)|h_t, i_t\right]. \quad (3)$$

This reward is defined using additional state-conditioned information available during training while remaining unbiased w.r.t. the standard history-based reward when taking expectations over $p(i_t|h_t)$ (cf. Lemma A.1 in Appendix A).

**Informed history $Q$-function.** We now introduce the informed history $Q$-function, which extends the history-state critic of Baisero & Amato (2022) to arbitrary state-conditioned privileged signals:

$$Q^\pi(h_t, i_t, a_t) = \mathbb{E}_{s_{t:\infty}, a_{t+1:\infty}}^\pi\left[G_t|h_t, i_t, a_t\right].$$

This $Q$-function is an unbiased estimator of the standard history-based counterpart (cf. Lemma A.2), i.e., $\mathbb{E}_{i_t|h_t}\left[Q^\pi(h_t, i_t, a_t)\right] = Q^\pi(h_t, a_t)$.

---

[1]This formulation is not restrictive. Suppose a POMDP with observation $o_t$ and auxiliary observation $o_t^+$ such that $(o_t, o_t^+) \sim O^+(o_t, o_t^+|s_t)$. Then, it is always possible to define an information $i_t = (o_t, o_t^+)$ such that $p(i_t, o_t|s_t) = I(i_t|s_t)\widetilde{O}(o_t|i_t)$ holds.

**Informed history value function.** Based on the proposed informed history $Q$-function, we can define the informed history value function:

$$V^\pi(h_t, i_t) = \mathbb{E}^\pi_{s_{t:\infty}, a_{t+1:\infty}}\big[G_t | h_t, i_t\big], \qquad (4)$$

which admits the time-invariant policy-based decomposition

$$V^\pi(h, i) = \sum_{a \in \mathcal{A}} \pi(a|h) \, Q^\pi(h, i, a). \qquad (5)$$

The corresponding Bellman equation is

$$Q^\pi(h, i, a) = R(h, i, a) + \gamma \, \mathbb{E}_{o', i'|h, i, a}\big[V^\pi(h', i')\big], \quad (6)$$

where $h' = hao'$. Analogously to the $Q$-function, the informed history value function is an unbiased estimator of the standard history value (cf. Lemma A.3), i.e., $\mathbb{E}_{i_t|h_t}\big[V^\pi(h_t, i_t)\big] = V^\pi(h_t)$.

In our setting, the privileged signal $i_t$ provides additional state-dependent context that can reduce the ambiguity about the underlying state given a fixed history. Formally, letting $H(s_t|h_t)$ denote the conditional entropy of the state, standard information-theoretic results imply $E_{i_t|h_t}\left[H(s_t|h_t, i_t)\right] \leq H(s_t|h_t)$, so conditioning on $i_t$ never increases, and on average reduces, uncertainty about the true state. This does not guarantee that $V^\pi(h_t, i_t)$ is inherently easier to approximate than $V^\pi(h_t)$; rather, it is the conditional expectation of a return random variable whose variance may be lower. In particular, by the law of total variance applied to the return $G_t$, $E_{i_t|h_t}\left[\text{Var}(G_t|h_t, i_t)\right] \leq \text{Var}(G_t|h_t)$, so an appropriate choice of $i_t$ can reduce the variance of value targets. It mainly occurs in environments with high "value aliasing", where distinct states yielding different returns may exist under the same observable history (Lambrechts et al., 2025). In such cases, $i_t$ can help disambiguate the state, providing a cleaner target for the critic.

When the privileged signal coincides with the full state, $i_t = s_t$, our formulation reduces to the history-state value function of Baisero & Amato (2022) (cf. Corollary B.1 in Appendix B), showing that their framework is a special case of our more general formalization.

The careful reader will have noticed that all definitions and unbiasedness results remain valid when replacing $i_t$ with an arbitrary auxiliary observation $o_t^+$. Indeed, since $h_t$ contains $o_t$ by definition, defining $\tilde{i}_t = (o_t, o_t^+)$ implies that conditioning on $(h_t, o_t^+)$ is equivalent to conditioning on $(h_t, \tilde{i}_t)$, so all results hold.

### 3.2. Informed Asymmetric Policy Gradient

Using the informed history-based $Q$-function, we define the informed asymmetric policy gradient

$$\nabla_\theta^{\text{IAAC}} J(\pi_\theta) = \mathbb{E}\bigg[\sum_{t=0}^\infty \gamma^t Q^\pi(h_t, i_t, a_t)\nabla_\theta \log \pi_\theta(a_t|h_t)\bigg],$$

where the policy $\pi_\theta$ depends only on the history $h_t$, while the critic may additionally condition on training-time information $i_t$.

Incorporating privileged state-conditioned information into the critic does not bias the policy gradient: for any choice of $i_t$, the informed asymmetric policy gradient coincides with the standard policy gradient.

**Theorem 3.1** (Informed asymmetric policy gradient). *Given an informed POMDP, the informed asymmetric policy gradient is equivalent to the standard policy gradient. Formally,*

$$\nabla_\theta^{\text{IAAC}} J(\pi_\theta) = \nabla_\theta J(\pi_\theta).$$

*Proof.* Given Equation 1 and Lemma A.2, we have

$$\nabla_\theta J(\pi_\theta) = \mathbb{E}\left[\sum_t \gamma^t Q^\pi(h_t, a_t) \nabla_\theta \underbrace{\log \pi_\theta(a_t|h_t)}_{=: \phi_t^{\pi_\theta}}\right]$$

$$\overset{(a)}{=} \sum_t \gamma^t \mathbb{E}_{h_t, a_t}[Q^\pi(h_t, a_t) \nabla_\theta \phi_t^{\pi_\theta}]$$

$$\overset{(b)}{=} \sum_t \gamma^t \mathbb{E}_{h_t, a_t}\big[\mathbb{E}_{i_t}[Q^\pi(h_t, i_t, a_t)|h_t] \nabla_\theta \phi_t^{\pi_\theta}\big]$$

$$\overset{(c)}{=} \sum_t \gamma^t \mathbb{E}_{h_t, i_t, a_t}[Q^\pi(h_t, i_t, a_t) \nabla_\theta \phi_t^{\pi_\theta}]$$

$$\overset{(d)}{=} \mathbb{E}\left[\sum_t \gamma^t Q^\pi(h_t, i_t, a_t) \nabla_\theta \log \pi_\theta(a_t|h_t)\right]$$

$$= \nabla_\theta^{\text{IAAC}} J(\pi_\theta).$$

Here, (a) and (d) follow from linearity of expectation and the assumption that the infinite discounted series is absolutely integrable, allowing interchange of expectation and summation; (b) uses Lemma A.2, and (c) applies the law of total expectation. This concludes the proof. $\square$

This result generalizes prior work on asymmetric actor-critic methods beyond state-based critics and recovers the asymmetric policy gradient of Baisero & Amato (2022) as a special case when $i_t = s_t$ (cf. Corollary B.2). Moreover, it extends straightforwardly to value functions that take an arbitrary auxiliary observation $o_t^+$ in place of $i_t$ as input.

Hence, Theorem 3.1 provides a theoretical justification for using arbitrary state-conditioned privileged signals during training. It shows that, under standard policy-gradient assumptions, symmetric and informed actor-critic methods optimize the same objective $J(\pi_\theta)$ and share the same first-order stationary condition $\nabla_\theta J(\pi_\theta) = 0$. Signal choice thus mainly affects optimization through changes in the variance of the value estimates, rather than asymptotic optimality.

In particular, our framework supports using a state-conditioned expert policy as a privileged signal, i.e., $i_t = a_t^\star \sim \pi^\star(\cdot|s_t)$, relating to ideas in asymmetric imitation learning but without introducing gradient bias. Indeed, $a_t^\star$ depends on the environment state $s_t$, and Theorem 3.1 guarantees unbiased policy gradients for the history-dependent actor, even when the expert policy itself is not deployable under partial observability. This enables the critic to exploit oracle information for value estimation without requiring the actor to perform direct imitation, which was shown to be suboptimal in asymmetric settings (Warrington et al., 2021).

**Informed history critic.**    As in symmetric actor-critic methods, we learn a value function using TD learning, and we use the TD error to form a low-variance advantage estimate $\hat{A}_t^{\text{IAAC}}$. The informed history critic $\hat{V}: \mathcal{H} \times \mathcal{I} \to \mathbb{R}$ additionally receives a privileged signal and approximates the informed history value $V^\pi(h_t, i_t)$. Combined with a history-dependent actor $\hat{\pi}_\theta(a_t|h_t)$, this yields the *informed asymmetric actor-critic (IAAC)*. The resulting policy gradient estimator is

$$\hat{\nabla}_\theta^{\text{IAAC}} J(\pi_\theta) = \mathbb{E}\left[\sum_t \gamma^t \, \hat{A}_t^{\text{IAAC}} \, \nabla_\theta \log \hat{\pi}_\theta(a_t|h_t)\right].$$

## 4. Informativeness of Privileged Signals

While the informed asymmetric actor-critic framework guarantees unbiased policy gradients for any state-conditioned privileged signals, not all such signals are equally useful for learning in practice. The choice of $i_t$ can affect the critic's learning efficiency and stability, motivating the need for criteria to select the most informative signals.

Here, we focus on the informativeness of $i_t \in \mathcal{I}$ from the perspective of predicting future returns, which is the critic's primary role. To this end, we introduce two criteria that quantify the informativeness of privileged signals and enable systematic comparison across different choices of $i_t$.

### 4.1. Residual Informativeness

The utility of a privileged signal $i_t$ in an informed critic depends on whether it provides information about future returns beyond what is already encoded in the observable history-action pair $(h_t, a_t)$. At the population level, this corresponds to the conditional independence (CI) hypothesis $\mathbb{H}_0^{\text{CI}} : G_t \perp\!\!\!\perp i_t | h_t, a_t$, which factorizes the joint distribution as $p(G_t, i_t, h_t, a_t) = p(h_t, a_t)p(G_t|h_t, a_t)p(i_t|h_t, a_t)$. Rejecting $\mathbb{H}_0^{\text{CI}}$ indicates that $i_t$ provides additional, non-redundant information about future returns.

One typically requires samples from both $\mathbb{H}_0^{\text{CI}}$ and $\mathbb{H}_1^{\text{CI}}$ to perform this test. Unfortunately, directly obtaining samples following $\mathbb{H}_0^{\text{CI}}$ is generally infeasible. Instead, interaction

with the environment yields samples from the unknown joint distribution $p(G_t, i_t, h_t, a_t) = p(h_t, a_t)p(G_t, i_t|h_t, a_t)$, where CI cannot be assumed. Sampling independently from $p(G_t|h_t, a_t)$ would require learning this conditional distribution, which is challenging for high-dimensional histories, and risks introducing noise into any direct test.

To circumvent this, we first remove the predictable components of $G_t$ and $i_t$ explained by $(h_t, a_t)$ and test for dependence in the residuals. As we explain below, testing residual independence provides insight into the independence of the original random variables, while simultaneously allowing sampling from a tractable approximation of $p(G_t|h_t, a_t)$.

**Definition 4.1** ($\alpha$-residual informativeness). Let $G_t$, $i_t$, $h_t$, and $a_t$ be random variables with finite second moments. Define the conditional-mean residuals

$$e_{G_t} := G_t - \mathbb{E}[G_t|h_t, a_t], \qquad e_{i_t} := i_t - \mathbb{E}[i_t|h_t, a_t].$$

A privileged signal $i_t$ is $\alpha$-*residual-informative* if a statistical test of independence rejects the null hypothesis

$$\mathbb{H}_0^{\text{res}} : e_{G_t} \perp\!\!\!\perp e_{i_t},$$

in favor of the alternative $\mathbb{H}_1^{\text{res}} : e_{G_t} \not\perp\!\!\!\perp e_{i_t}$, at significance level $\alpha > 0$.

CI implies residual independence, but the converse need not hold in general (Zhang et al., 2023). Hence, the residual test provides a necessary, but not sufficient, condition for CI, which aligns with our goal of detecting whether $i_t$ carries additional predictive information about $G_t$ beyond $(h_t, a_t)$. This reduces the problem to testing whether the unexplained components of $G_t$ and $i_t$ remain statistically dependent.

Testing $\mathbb{H}_0^{\text{res}}$ ideally requires samples from

$$p(e_{G_t}, e_{i_t}, h_t, a_t) = p(e_{G_t}|h_t, a_t) \, p(e_{i_t}|h_t, a_t) \, p(h_t, a_t),$$

which, as with the original random variables, is generally intractable. To approximate a sample of $G_t$ independent of $i_t$ but with the correct conditional mean, we construct a surrogate $\tilde{G}_t^{\text{null}}$ in two steps: (i) we an independent sample $\tilde{G}_t$ from the marginal distribution $p(G_t)$ of $G_t$, which is easily feasible from the collected episodes; (ii) we shift these samples to preserve the estimated conditional mean $\mathbb{E}[G_t|h_t, a_t]$. Thus, $\tilde{G}_t^{\text{null}} = \tilde{G}_t - \mathbb{E}[\tilde{G}_t] + \mathbb{E}[G_t|h_t, a_t]$.

By construction, the distribution of this sample exactly matches the first moment of the target distribution $p(G_t|h_t, a_t)$, while its higher-order moments follow those of the marginal distribution $p(G_t)$.

The corresponding null residual

$$e_{\tilde{G}_t}^{\text{null}} := \tilde{G}_t^{\text{null}} - \mathbb{E}[G_t|h_t, a_t] = \tilde{G}_t - \mathbb{E}[\tilde{G}_t]$$

is independent of $e_{i_t}$, and thus satisfies $\mathbb{H}_0^{\text{res}}$. Further details on the derivation of the null residual are given in Appendix C.

**Algorithm 1** $\alpha$-Residual-Informativeness Test

---

**Input:** Dependency measure $\rho(\cdot, \cdot)$, number of folds $K$, independently collected episodes $\{(o_t, a_t, i_t, G_t)\}_{t=0}^{T-1}$, number of permutations $B$, significance level $\alpha$.

Encode histories with recurrent network: $z_t = f_{\text{RNN}}(h_t)$.

Partition data into $K$ folds.

**for** each fold $k = 1$ to $K$ **do**

    Train regression models on $K-1$ folds:

    Predict $\hat{\mathbb{E}}[G_t|z_t, a_t]$ and $\hat{\mathbb{E}}[i_t|z_t, a_t]$ on held-out fold.

    Compute residuals $\hat{e}_{G_t}$ and $\hat{e}_{i_t}$ on held-out fold (Eq. 7).

**end for**

Aggregate cross-fitted residuals across held-out folds.

Compute observed dependence $\rho_{\text{obs}}$ using $\rho(\cdot, \cdot)$.

**for** $b = 1$ to $B$ **do**

    Permute residuals $\hat{e}_{G_t}$ on episode-level.

    Compute $\rho(\hat{e}_{G_t}^{(b)}, \hat{e}_{i_t})$ for permuted residuals.

**end for**

Compute empirical $p$-value according to Eq. 8.

**Output:** Declare $i_t$ $\alpha$-residual-informative if $p < \alpha$.

---

**Practical Implementation.** In practice, we approximate $\mathbb{E}[G_t|h_t, a_t]$ and $\mathbb{E}[i_t|h_t, a_t]$ using regression models capable of capturing nonlinear relationships (e.g., random forests or neural networks), since $G_t$ and $i_t$ typically exhibit nonlinear dependence. Histories are generally encoded using a recurrent network to produce a fixed-length representation $z_t = f_{\text{RNN}}(h_t)$, yielding residual estimates

$$\hat{e}_{G_t} := G_t - \hat{\mathbb{E}}[G_t|z_t, a_t], \quad \hat{e}_{i_t} := i_t - \hat{\mathbb{E}}[i_t|z_t, a_t], \quad (7)$$

and for the null sample, we have $\hat{e}_{\tilde{G}_t}^{\text{null}} := \tilde{G}_t^{\text{null}} - \hat{\mathbb{E}}[G_t|z_t, a_t]$. We quantify the dependence between $\hat{e}_{G_t}$ and $\hat{e}_{i_t}$ using a general dependency measure $\rho(\cdot, \cdot)$ (e.g., mutual information or Hilbert-Schmidt Independence Criterion). By design, $\rho(\hat{e}_{G_t}, \hat{e}_{i_t}) = 0$ under independence and increases with stronger dependence. To account for finite-sample variability and temporal correlations, we approximate the surrogate null by episode-wise permutation of the observed residuals. Residuals $\hat{e}_{G_t}$ are shuffled across independently collected episodes, while keeping the temporal ordering within each episode unchanged, thereby preserving within-episode dependence structure and the estimated conditional mean. Cross-fitting is used to prevent overfitting: the data set is split into $K$ folds, with regressors trained on $K-1$ folds and evaluated on the held-out fold.

We compute the empirical $p$-value with $B$ permutations as

$$p = \frac{1 + \sum_{b=1}^{B} \mathbf{1}\{\rho(\hat{e}_{G_t}^{(b)}, \hat{e}_{i_t}) \geq \rho(\hat{e}_{G_t}, \hat{e}_{i_t})\}}{B + 1}, \quad (8)$$

where $\hat{e}_{G_t}^{(b)}$ denotes the residual under the $b$-th permutation and $\mathbf{1}\{\cdot\}$ is the indicator function, which equals 1 if the

condition inside the braces holds and 0 otherwise. The privileged signal $i_t$ is declared $\alpha$-residual-informative if $p < \alpha$. Algorithm 1 summarizes the procedure.

Notably, the $\alpha$-residual-informativeness criterion can be evaluated on episodes collected under any exploratory policy, without requiring a trained actor or critic. In particular, a random policy can be used, although coverage of the state-action space may be limited. This allows estimating the informativeness of privileged signals even before training and supports informed decisions about which signals to include. To obtain meaningful residuals, the history encoding function $f_{\text{RNN}}(\cdot)$ is typically trained in advance to produce effective representations of the interaction history. More generally, any encoding scheme (e.g., fixed-window or padding-based approaches) can be used, as long as it maps the history to a fixed-length representation.

As detailed in Appendix C.2, for Lipschitz-continuous dependency measures, perturbations in the residuals can be bounded in terms of the regression error. If this error does not decrease with increasing sample size, the resulting test may deviate from its nominal significance level and exhibit reduced statistical power. To address this, we use cross-fitting and random forests, which are typically less prone to overfitting than deep neural networks in low-data regimes.

### 4.2. Informativeness via Return Prediction Gain

Informativeness can also be assessed via the critic's predictive accuracy: a privileged signal is informative if its inclusion improves value estimation. This post-hoc criterion provides a quantitative measure of a signal's contribution to return prediction. It requires only episodes collected under any fixed policy and does not depend on policy performance.

Consider a symmetric critic $\hat{Q}(h_t, a_t)$ and an informed asymmetric critic $\hat{Q}(h_t, i_t, a_t)$. For a set of $N$ episodes $\{\tau_j\}_{j=1}^{N}$, we define the episode-level squared-error gain

$$L^{\tau_j} := \frac{1}{T_j} \sum_{t=0}^{T_j-1} \left( (\hat{Q}(h_t, a_t) - G_t)^2 - (\hat{Q}(h_t, i_t, a_t) - G_t)^2 \right),$$

where $T_j$ is the length of episode $\tau_j$. $L^{\tau_j} > 0$ indicates that including $i_t$ improves return prediction on that episode.

**Definition 4.2** (($\epsilon, \delta$)-prediction informativeness). A privileged signal $i_t$ is said to be ($\epsilon, \delta$)-*prediction-informative* with $\epsilon \geq 0$ if a statistical test rejects the null hypothesis

$$\mathbb{H}_0 : \mathbb{E}[L^\tau] \leq \epsilon,$$

in favor of the alternative $\mathbb{H}_1 : \mathbb{E}[L^\tau] > \epsilon$, at significance level $\delta$, based on the set of episode-level gains $\{L^{\tau_j}\}_{j=1}^{N}$.

**Practical Implementation.** In practice, the test procedure depends on the number of episodes collected $N$. For small

sample sizes, we apply a one-sided bootstrap test: we resample $B > N$ episodes with replacement, compute bootstrap means $\bar{L}_b^*$ for $1 \leq b \leq B$, and estimate the empirical $p$-value as $p = \frac{1+\sum_{b=1}^{B} \mathbf{1}\{\bar{L}_b^* \leq 0\}}{B+1}$, rejecting $\mathbb{H}_0$ if $p < \delta$. For larger sample sizes (e.g., $N > 1{,}000$), we apply a one-sided $t$-test based on the sample mean $\bar{L}$ and variance $\hat{\sigma}^2$. The test statistic is $t = \frac{\bar{L}-\epsilon}{\hat{\sigma}/\sqrt{N}}$, which under $\mathbb{H}_0$ follows a Student-$t$ distribution with $N-1$ degrees of freedom. We reject $\mathbb{H}_0$ if the corresponding one-sided $p$-value is below $\delta$.

### 4.3. Selecting Privileged Signal Generators

Typically, we consider a state-conditioned candidate feature vector $c_t = (c_t^1, \ldots, c_t^M)$, with $M \in \mathbb{Z}_{\geq 0}$, where each component corresponds to a candidate feature extracted from the underlying state or auxiliary training-time information. A privileged signal is then formed by selecting a subset of components $\mathcal{Z} \subseteq \{1, \ldots, M\}$, yielding the signal $i_t := (c_t^m)_{m \in \mathcal{Z}}$. Different subsets may encode distinct aspects of the environment state and are not assumed to be equally informative a priori.

Both proposed informativeness criteria support principled selection over such feature subsets via hypothesis testing. For all subsets, we evaluate the sequence $\{(c_t^m)_{m \in \mathcal{Z}}\}_{t=0}^T$ and deem the feature subset informative if the corresponding null hypothesis is rejected. In the pre-training setting, we apply the $\alpha$-residual-informativeness test and retain subsets for which residual independence is rejected at significance level $\alpha$. In the post-hoc setting, we apply the $(\epsilon, \delta)$-prediction-informativeness test and retain subsets that yield a statistically significant improvement in value prediction accuracy ($\epsilon = 0$). Among informative candidates, one may select the feature subset with the largest effect size or smallest $p$-value, corresponding to the privileged signal that provides the strongest additional information about future returns under the respective criterion.

When only a single $i_t$ is available or when $c_t$ cannot be naturally decomposed into $M$ components, the same procedure can still be used to compare different learned encodings of the signal, or of the observation in the degenerate case $i_t = o_t$. In all cases, Theorem 3.1 applies without assumptions on the dimensionality or structure of $i_t$.

## 5. Experiments

We evaluate our informed asymmetric actor-critic framework along two dimensions. First, we assess performance on standard POMDP benchmarks against symmetric and asymmetric baselines. Second, we validate the proposed informativeness criteria in controlled informed POMDPs, where privileged signals can be systematically varied. We

release our source code on GitHub[2] to ensure reproducibility and report wall-clock runtimes in Appendix F. Ablations and further results can be found in Appendix G.

### 5.1. Benchmark Performance

**Environments.** We use six benchmark navigation tasks from Baisero & Amato (2022) and six environments from the POPGym suite (Morad et al., 2023). The navigation tasks consist of *Heaven-Hell-3*, *Shopping-5*, *Car-Flag*, *Cleaner*, *Memory-Four-Rooms-7x7*, and *Memory-Four-Rooms-9x9*, while the POPGym suite includes *Higher Lower*, *Repeat First*, *Repeat Previous*, *Count Recall*, *Concentration*, and *Position Cart Pole*. For each environment, we define task-specific privileged signals available only to the critic; all policies receive identical inputs without access to additional information. Detailed descriptions of all environments are provided in Appendix D.1.

**Baselines.** We compare our informed asymmetric actor-critic method, *informed-asym-A2C*, against three advantage actor-critic (A2C) variants: (1) *A2C*, a symmetric approach using a history-based critic $\hat{V}(h)$; (2) *asym-A2C-hs*, an asymmetric variant with a history-state critic $\hat{V}(h, s)$; and (3) *asym-A2C-s*, an asymmetric approach with a state-based critic $\hat{V}(s)$, evaluated only on the navigation tasks. We use model architectures and hyperparameters from Baisero & Amato (2022) and Morad et al. (2023), respectively; see Appendix E.1.

**Results.** Figure 1 reports the learning curves across all twelve benchmark tasks, showing episodic returns smoothed over 100 episodes and aggregated over 20 independent runs. For the POPGym environments, we present only the best-performing variant of *informed-asym-A2C*. Across navigation environments, our method consistently improves sample efficiency and training stability relative to *A2C*. In *Heaven-Hell-3*, *informed-asym-A2C* exhibits strong performance relative to *A2C* and *asym-A2C-s*, though it is slightly outperformed by *asym-A2C-hs*. In *Shopping-5*, the informed asymmetric actor-critic converges faster than *asym-A2C-hs* and achieves comparable asymptotic return. In *Car-Flag*, it outperforms all baselines in both convergence speed and asymptotic return. In *Cleaner*, *asym-A2C-hs* achieves marginally higher returns, but *informed-asym-A2C* converges at a similar rate with greater stability than *A2C*, which suffers a performance drop after 2.5 million steps. In the Memory-Four-Rooms tasks, *informed-asym-A2C* outperforms both asymmetric baselines. These findings are consistent with those in the POPGym environments. *Informed-asym-A2C* achieves the highest final returns in most of the environments. In *Higher Lower*, all methods perform similarly. On *Repeat First*, *A2C* peforms best,

---

[2] https://github.com/EbiDa/informed-asymmetric-a2c

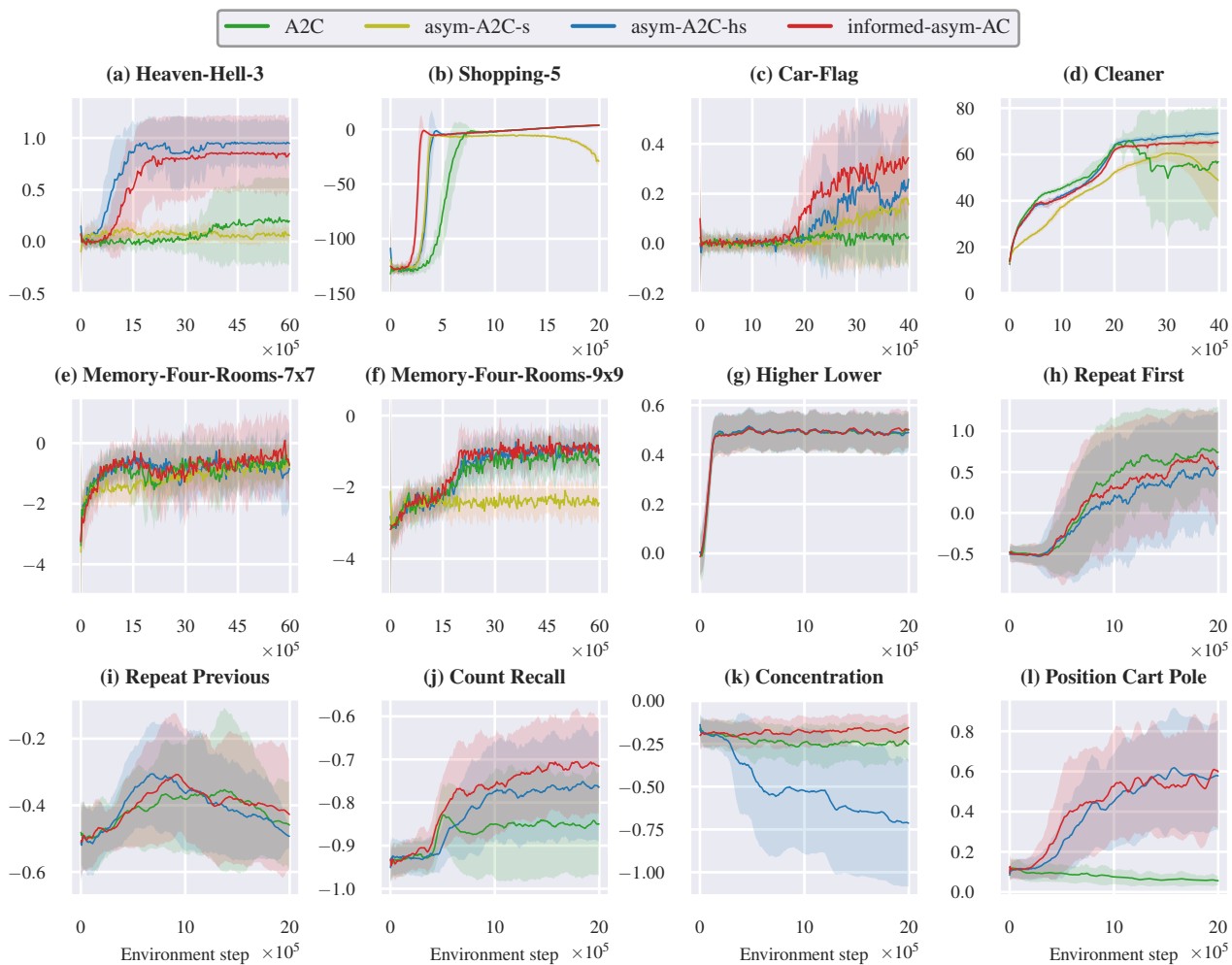

*Figure 1.* Learning performance on six benchmark navigation tasks (a-f) and six POPGym environments (g-l). Curves show mean episodic returns smoothed using a moving average over 100 episodes. Means and standard deviations are computed across 20 independent runs

while on *Position Cart Pole* both asymmetric variants clearly outperform *A2C*, which fails to converge within 2 million environment steps. For *Concentration*, the full-state A2C variant appears to struggle with convergence, likely due to the high-dimensional state representation with many potentially irrelevant features.

Overall, these results demonstrate that asymmetric critics leveraging appropriately structured privileged signals can match or outperform full-state critics while relying on strictly less information.

### 5.2. Validation of Informativeness Criteria

**Environments.** To empirically validate the proposed informativeness criteria in a controlled setting, we consider a distribution of synthetic informed POMDPs with finite state space ($|\mathcal{S}| = 20$), discrete action space ($|\mathcal{A}| = 4$), and continuous observation and privileged information spaces.

Transition dynamics are generated following François-Lavet et al. (2019) by sparsifying state-action transitions and normalizing to obtain valid probability distributions. Each state is associated with a latent Gaussian feature vector ($s_t \in \mathbb{R}^5$), and rewards are defined as linear functions of these features with weights $w_r$ sampled uniformly from $[-1, 1]$. Privileged signals $i_t$ are constructed by masking subsets of state features, while observations $o_t$ correspond to noisy feature subsets of $i_t$; see Appendix D.3 for details.

**Informativeness criteria.** For the residual-based criterion, we estimate dependence between return-prediction residuals and privileged-signal residuals via the Hilbert-Schmidt Independence Criterion (HSIC) (Gretton et al., 2005) with Gaussian RBF kernels, where bandwidths are chosen via the median heuristic and a Nyström approximation (Kalinke & Szabó, 2023) with 512 landmarks is applied for efficiency. We collect 250 episodes of length 25 under a random

*Table 1.* Comparison of candidate privileged signals using the residual-based and post-hoc prediction informativeness criteria. We provide mean $\pm$ standard deviation across ten independent runs on a randomly sampled informed POMDP with reward weights $w_r = [0.0001, 0.0001, -0.0001, -1.0, 1.0]$. We further include the area under the mean episodic return curve (AUC), computed from 50 test episodes evaluated every 50 gradient steps during training. The symmetric A2C baseline achieves an AUC of $1.06 \times 10^5 \pm 1.61 \times 10^4$.

| PRIVILEGED SIGNAL | RESIDUAL INFORMATIVENESS | | PREDICTION INFORMATIVENESS | | AUC |
|---|---|---|---|---|---|
| | $\rho_{\mathrm{obs}}$ | $p$-value | $L_\tau$ | $p$-value | (sum of returns) |
| $i_t = [s_t^1, s_t^2]$ | 3.0e-05 $\pm$ 8.1e-06 | 0.438 $\pm$ 0.290 | -1.4e-02 $\pm$ 0.014 | 0.801 $\pm$ 0.330 | 1.07e+05 $\pm$ 1.58e+04 |
| $i_t = [s_t^1, s_t^2, s_t^3]$ | 5.5e-05 $\pm$ 1.9e-05 | 0.170 $\pm$ 0.169 | 0.007 $\pm$ 0.016 | 0.397 $\pm$ 0.338 | 1.08e+05 $\pm$ 1.56e+04 |
| $i_t = [s_t^1, s_t^2, s_t^4]$ | 7.6e-05 $\pm$ 3.2e-05 | 0.119 $\pm$ 0.262 | 0.035 $\pm$ 0.018 | 0.043 $\pm$ 0.077 | 1.08e+05 $\pm$ 1.57e+04 |
| $i_t = [s_t^1, s_t^2, s_t^5]$ | 5.0e-05 $\pm$ 1.5e-05 | 0.191 $\pm$ 0.228 | 0.018 $\pm$ 0.019 | 0.232 $\pm$ 0.248 | 1.10e+05 $\pm$ 1.69e+04 |
| $i_t = [s_t^1, s_t^2, s_t^3, s_t^4]$ | 7.4e-05 $\pm$ 2.4e-05 | 0.068 $\pm$ 0.116 | 0.046 $\pm$ 0.018 | 0.009 $\pm$ 0.016 | 1.07e+05 $\pm$ 1.56e+04 |
| $i_t = [s_t^1, s_t^2, s_t^3, s_t^5]$ | 5.8e-05 $\pm$ 1.5e-05 | 0.106 $\pm$ 0.141 | 0.026 $\pm$ 0.019 | 0.120 $\pm$ 0.152 | 1.11e+05 $\pm$ 1.97e+04 |
| $i_t = [s_t^1, s_t^2, s_t^4, s_t^5]$ | 7.6e-05 $\pm$ 2.1e-05 | 0.035 $\pm$ 0.051 | 0.064 $\pm$ 0.020 | 0.003 $\pm$ 0.006 | 1.23e+05 $\pm$ 1.76e+04 |
| $i_t = [s_t^1, s_t^2, s_t^3, s_t^4, s_t^5]$ | 7.0e-05 $\pm$ 1.8e-05 | 0.038 $\pm$ 0.051 | 0.056 $\pm$ 0.032 | 0.072 $\pm$ 0.131 | 1.19e+05 $\pm$ 1.83e+04 |

policy and perform episode-level permutation testing with $B = 1{,}000$ resamples. Conditional expectations are estimated via cross-fitting with a 100-tree random forest using 5-fold cross-validation. Observation-action histories are encoded with an RNN of width 64, trained on 100 training episodes. For the post-hoc criterion, we train symmetric and informed recurrent critics via TD learning using 5-fold cross-validation on the same environments for 2,500 episodes of length 25. In the informed critic, the 64-dimensional representation of the history is concatenated with the privileged signal and passed through a feedforward network to output a single value estimate. We evaluate episode-level gains in value prediction accuracy across held-out folds, with $\epsilon = 0$.

**Results.** Table 1 reports test statistics and $p$-values for different candidate privileged signals under both criteria across ten independent runs on a randomly sampled environment. In this setting, the agent observes noiseless $o_t = (s_t^1, s_t^2)$, while the reward weight vector $w_r$ assigns most of its weight to state components $s_t^4$ and $s_t^5$, which therefore primarily determine the reward. Privileged signals that include these components consistently exhibit stronger residual dependence and higher predictive gains. In contrast, signals containing only redundant or irrelevant components (e.g., $s_t^3$) yield weaker statistical evidence under both criteria. Notably, full-state access does not always achieve the best scores, indicating that additional features can be uninformative when misaligned with return prediction. Overall, the results show that both criteria effectively identify informative signals and that performance (cf. AUC in Table 1) is driven by selecting return-relevant features.

### 5.3. Discussion

Our findings suggest that learning performance depends primarily on whether the input contains reward-relevant information, rather than on the amount of information provided. In particular, adding state features that are unrelated

or only weakly related to future returns can degrade value estimation by introducing structured noise, even when using high-capacity function approximators. This effect is especially pronounced when inputs include rapidly varying components unrelated to the return signal, akin to a "noisy TV" scenario.

Notably, selecting informative training-time signals is inherently task-dependent, similar in spirit to the design of observation spaces. Our framework extends this perspective by treating training-time information as an additional degree of freedom, allowing auxiliary variables to serve as candidate signals. Our criteria automatically identify informative subsets from a large set of candidates in a fully data-driven manner, without requiring manual feature engineering. This motivates principled signal selection as a complementary factor to architecture design and optimization in asymmetric actor-critic methods.

## 6. Conclusion

We introduced the informed asymmetric actor-critic framework, which enables the critic to leverage arbitrary state-dependent privileged signals during training while preserving unbiased policy gradients. This generalizes asymmetric actor-critic methods beyond full-state critics and shifts the focus from access to state information towards the choice of informative training-time signals. We further proposed two complementary criteria to assess signal informativeness, and demonstrated empirically that appropriately selected signals can improve value estimation and learning efficiency.

Future work may extend these criteria to account for signal complexity or direct policy effects, enabling better trade-offs between informativeness and model capacity, integrate signal selection end-to-end within the learning loop, and explore asymmetric critics that condition on histories of privileged signals.

## Acknowledgments

This work was supported by the *Helmholtz Association Initiative and Networking Fund* on the HAICORE@KIT partition. Daniel Ebi acknowledges financial support by the *German Research Foundation (DFG)* as part of the Research Training Group GRK 2153: "Energy Status Data – Informatics Methods for its Collection, Analysis and Exploitation". This work was carried out while Gaspard Lambrechts was a postdoctoral fellow of the Fund for Scientific Research (FNRS) at the University of Liège, supported by the *Wallonia-Brussels Federation* in Belgium.

## Impact Statement

This paper presents work whose goal is to advance the field of Machine Learning. There are many potential societal consequences of our work, none which we feel must be specifically highlighted here.

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

# A. Unbiasedness Results

This section establishes unbiasedness properties of the informed history-based reward (Lemma A.1), $Q$- (Lemma A.2), and value functions (Lemma A.3), which are central to the informed asymmetric actor-critic framework.

**Lemma A.1** (Unbiasedness of the informed history-based reward). *In an informed POMDP, the informed history-based reward function $R(h_t, i_t, a_t)$ satisfies*

$$\mathbb{E}_{i_t|h_t}[R(h_t, i_t, a_t)] = R(h_t, a_t),$$

*for all $h_t \in \mathcal{H}$ and $a_t \in \mathcal{A}$, where the expectation is taken under the belief $p(i_t|h_t)$.*

*Proof.* Using the definition of the standard history-based reward function, i.e.,

$$R(h_t, a_t) = \mathbb{E}_{s_t|h_t}[R(s_t, a_t)] = \sum_{s_t \in \mathcal{S}} R(s_t, a_t)\, p(s_t|h_t),$$

and applying the law of total probability, we obtain:

$$
\begin{aligned}
R(h_t, a_t) &= \sum_{s_t \in \mathcal{S}} p(s_t|h_t) R(s_t, a_t) \\
&= \sum_{s_t \in \mathcal{S}} \Big( \sum_{i_t \in \mathcal{I}} p(s_t|h_t, i_t)\, p(i_t|h_t) \Big) R(s_t, a_t) \\
&= \sum_{i_t \in \mathcal{I}} \Big( \sum_{s_t \in \mathcal{S}} R(s_t, a_t)\, p(s_t|h_t, i_t) \Big) p(i_t|h_t) \\
&= \mathbb{E}_{i_t|h_t}\Big[ \mathbb{E}_{s_t|h_t, i_t}\big[ R(s_t, a_t) \big] \Big] \\
&= \mathbb{E}_{i_t|h_t}\big[ R(h_t, i_t, a_t) \big].
\end{aligned}
$$

This concludes the proof. $\qquad\square$

**Lemma A.2** (Unbiasedness of the informed history $Q$-function). *In an informed POMDP, the informed history $Q$-function $Q^\pi(h_t, i_t, a_t)$ satisfies*

$$\mathbb{E}_{i_t|h_t}[Q^\pi(h_t, i_t, a_t)] = Q^\pi(h_t, a_t),$$

*for all $h_t \in \mathcal{H}$ and $a_t \in \mathcal{A}$.*

*Proof.* Starting with the definition of the history $Q$-function and using the law of total expectation, we have:

$$
\begin{aligned}
Q^\pi(h_t, a_t) &= \mathbb{E}^\pi_{s_{t:\infty}, a_{t+1:\infty}}\left[ \sum_{k=0}^{\infty} \gamma^k\, R\big(s_{t+k}, a_{t+k}\big) \Big| h_t, a_t \right] \\
&= \mathbb{E}_{i_t|h_t}\left[ \mathbb{E}^\pi_{s_{t:\infty}, a_{t+1:\infty}}\left[ \sum_{k=0}^{\infty} \gamma^k\, R\big(s_{k+t}, a_{k+t}\big) \Big| h_t, i_t, a_t \right] \right] \\
&= \mathbb{E}_{i_t|h_t}\left[ Q^\pi(h_t, i_t, a_t) \right].
\end{aligned}
$$

This concludes the proof. $\qquad\square$

**Lemma A.3** (Unbiasedness of the informed history value function). *In an informed POMDP, the informed history value function $V^\pi(h_t, i_t)$ satisfies for all $h_t \in \mathcal{H}$:*

$$\mathbb{E}_{i_t|h_t}[V^\pi(h_t, i_t)] = V^\pi(h_t).$$

*Proof.* Given the definition of the history value function, i.e.,

$$V^\pi(h_t) = \mathbb{E}^\pi_{s_{t:\infty},\, a_{t+1:\infty}}\left[ \sum_{k=0}^{\infty} \gamma^k R(s_{k+t}, a_{k+t}) \Big| h_t \right],$$

and using the law of total expectation, we have:

$$V^\pi(h_t) = \mathbb{E}^\pi_{s_{t:\infty}, a_{t+1:\infty}} \left[ \sum_k^\infty \gamma^k R(s_{k+t}, a_{k+t}) \Big| h_t \right]$$

$$= \mathbb{E}_{i_t|h_t} \left[ \mathbb{E}^\pi_{s_{t:\infty}, a_{t+1:\infty}} \left[ \sum_k^\infty \gamma^k R(s_{k+t}, a_{k+t}) \Big| h_t, i_t \right] \right]$$

$$= \mathbb{E}_{i_t|h_t} \left[ V^\pi(h_t, i_t) \right].$$

This concludes the proof. $\qquad\square$

## B. Auxiliary Results

This section collects our auxiliary results.

**Corollary B.1** (Relation of $V^\pi(h, i)$ to the history-state value function of Baisero & Amato (2022))**.** *The informed history value function $V^\pi(h, i)$ reduces to the history-state value function for $i = s$, where $s \in \mathcal{S}$ denotes the true environment state. In particular,*

$$V^\pi(h, s) = \sum_{a \in \mathcal{A}} \pi(a|h) Q^\pi(h, s, a),$$

*where the history-state action-value function is defined as*

$$Q^\pi(h, s, a) = R(s, a) + \gamma E_{s', o'} \left[ V^\pi(h', s') | s, a \right],$$

*with $s' \sim p(s'|s, a)$, $o' \sim \widetilde{O}(o'|s')$, $i' = s'$, and $h'$ denoting the updated history resulting from appending action $a$ and observation $o'$ to $h$.*

*By Lemma A.3, this formulation provides an alternative unbiased estimator of the history value function:*

$$V^\pi(h) = E_{s|h} \left[ V^\pi(h, s) \right],$$

*as previously established by Baisero & Amato (2022).*

**Corollary B.2** (Relation of $\nabla^{\mathrm{IAAC}}_\theta J(\pi_\theta)$ to the asymmetric policy gradient of Baisero & Amato (2022))**.** *The informed asymmetric policy gradient $\nabla^{I\!AAC}_\theta J(\pi_\theta)$ reduces to the asymmetric policy gradient introduced by Baisero & Amato (2022) for $i = s$, where $s \in \mathcal{S}$ denotes the true environment state. In particular,*

$$\nabla^{AC}_\theta J(\pi_\theta) = E \left[ \sum_{t=0}^\infty \gamma^t Q^\pi(h_t, s_t, a_t) \nabla_\theta \log \pi_\theta(a_t|h_t) \right].$$

*Following Lemma A.1 and A.3, this formulation recovers an alternative asymmetric policy gradient estimator that is equivalent to the standard policy gradient:*

$$\nabla^{AC}_\theta J(\pi_\theta) = \nabla_\theta J(\pi_\theta),$$

*as established by Baisero & Amato (2022).*

## C. Analysis of the $\alpha$-Residual-Informativeness Test

In this section, we describe the construction of the proposed $\alpha$-residual-informativeness test and show how regression error affects its statistical power.

### C.1. Test Construction

First, we formalize the null distribution implicitly approximated by the permutation procedure used in the residual-based informativeness test. Our goal is to clarify how the proposed test relates to the conditional independence (CI) hypothesis

$$\mathbb{H}^{\mathrm{CI}}_0 : \quad G_t \perp\!\!\!\perp i_t | h_t, a_t.$$

Let $G_t, i_t, h_t$, and $a_t$ be random variables with joint distribution $p(G_t, i_t, h_t, a_t)$ and finite second moments. Then, the conditional-mean residuals are defined as

$$e_{G_t} := G_t - \mathbb{E}[G_t|h_t, a_t], \qquad e_{i_t} := i_t - \mathbb{E}[i_t|h_t, a_t].$$

**Proposition C.1.** *If* $\mathbb{H}_0^{\mathrm{CI}}$ *holds, then*

$$e_{G_t} \perp\!\!\!\perp e_{i_t} \mid h_t, a_t.$$

*Proof.* Under $\mathbb{H}_0^{\mathrm{CI}}$, the conditional joint distribution factorizes:

$$p(G_t, i_t|h_t, a_t) = p(G_t|h_t, a_t)\, p(i_t|h_t, a_t).$$

Subtracting measurable functions of $(h_t, a_t)$ from each variable does not affect CI given $(h_t, a_t)$, so

$$p(e_{G_t}, e_{i_t}|h_t, a_t) = p(e_{G_t}|h_t, a_t)\, p(e_{i_t}|h_t, a_t).$$

This concludes the proof. $\qquad\square$

The converse does not hold in general (Zhang et al., 2023); thus, testing residual independence is a relaxation of CI testing.

**Ideal residual null distribution.** An exact test of $\mathbb{H}_0^{\mathrm{res}} : e_{G_t} \perp\!\!\!\perp e_{i_t} \mid h_t, a_t$ would require samples from

$$p(e_{G_t}, e_{i_t}, h_t, a_t) = p(h_t, a_t)\, p(e_{G_t}|h_t, a_t)\, p(e_{i_t}|h_t, a_t).$$

While $(h_t, a_t, e_{i_t})$ can be sampled from data once $\mathbb{E}[i_t|h_t, a_t]$ is estimated, obtaining independent samples from $p(e_{G_t}|h_t, a_t)$ is intractable because it requires sampling from the unknown return distribution conditioned on the full history $h_t$ and action $a_t$.

**First-moment-matching surrogate.** To construct a tractable null distribution, we replace the original return $G_t$ by a surrogate variable.

**Assumption C.2** (Existence of a surrogate). There exists a random variable $\tilde{G}_t$ defined on the same probability space such that

(i) $\tilde{G}_t \overset{d}{=} G_t$ (same marginal distribution),

(ii) $\tilde{G}_t \perp\!\!\!\perp i_t, h_t, a_t$ (independence from the privileged signal, history, and action),

(iii) $\mathbb{E}[\tilde{G}_t^2] < \infty$ (finite second moment).

Under Assumption C.2, the surrogate null $\tilde{G}_t^{\mathrm{null}}$ is then obtained by replacing $G_t$ with $\tilde{G}_t$, i.e.,

$$\tilde{G}_t^{\mathrm{null}} := \tilde{G}_t - \mathbb{E}[\tilde{G}_t] + \mathbb{E}[G_t|h_t, a_t]. \qquad (9)$$

The corresponding surrogate null residual is

$$e_{\tilde{G}_t}^{\mathrm{null}} := \tilde{G}_t^{\mathrm{null}} - \mathbb{E}[G_t|h_t, a_t] = \tilde{G}_t - \mathbb{E}[\tilde{G}_t].$$

**Proposition C.3** (Properties of the surrogate residual). *For all* $(h_t, a_t)$,

$$\mathbb{E}[e_{\tilde{G}_t}^{null}|h_t, a_t] = 0, \qquad \text{and} \qquad e_{\tilde{G}_t}^{null} \perp\!\!\!\perp i_t|h_t, a_t.$$

*Proof.* First, we prove zero conditional mean. By definition,

$$e_{\tilde{G}_t}^{\mathrm{null}} = \tilde{G}_t - \mathbb{E}[\tilde{G}_t],$$

where $\mathbb{E}[\tilde{G}_t]$ is a constant. Taking the conditional expectation given $(h_t, a_t)$, yields

$$\mathbb{E}[e_{\tilde{G}_t}^{\text{null}}|h_t, a_t] = \mathbb{E}[\tilde{G}_t|h_t, a_t] - \mathbb{E}[\tilde{G}_t].$$

Since $\tilde{G}_t$ is independent of $(h_t, a_t)$ by Assumption C.2(ii),

$$\mathbb{E}[\tilde{G}_t|h_t, a_t] = \mathbb{E}[\tilde{G}_t].$$

Therefore,

$$\mathbb{E}[e_{\tilde{G}_t}^{\text{null}}|h_t, a_t] = 0.$$

Second, we show conditional independence from $i_t$. Again using

$$e_{\tilde{G}_t}^{\text{null}} = \tilde{G}_t - \mathbb{E}[\tilde{G}_t],$$

note that subtracting a constant does not affect independence relations. Thus, it suffices to show

$$\tilde{G}_t \perp\!\!\!\perp i_t | h_t, a_t.$$

For any measurable sets $\mathcal{X}, \mathcal{Y}$,

$$p(\tilde{G}_t \in \mathcal{X}, i_t \in \mathcal{Y}|h_t, a_t) = \frac{p(\tilde{G}_t \in \mathcal{X}, i_t \in \mathcal{Y}, h_t, a_t)}{p(h_t, a_t)}. \tag{10}$$

Since $\tilde{G}_t$ is independent of $(h_t, a_t, i_t)$,

$$p(\tilde{G}_t \in \mathcal{X}, i_t \in \mathcal{Y}, h_t, a_t) = p(\tilde{G}_t \in \mathcal{X})\, p(i_t \in \mathcal{Y}, h_t, a_t).$$

Substituting into Equation 10 gives

$$p(\tilde{G}_t \in \mathcal{X}, i_t \in \mathcal{Y}|h_t, a_t) = p(\tilde{G}_t \in \mathcal{X})\, \frac{p(i_t \in \mathcal{Y}, h_t, a_t)}{p(h_t, a_t)}$$
$$= p(\tilde{G}_t \in \mathcal{X})\, p(i_t \in \mathcal{Y}|h_t, a_t).$$

Finally, because $\tilde{G}_t \perp\!\!\!\perp h_t, a_t$, we have

$$p(\tilde{G}_t \in \mathcal{X}) = p(\tilde{G}_t \in \mathcal{X}|h_t, a_t).$$

Therefore,

$$p(\tilde{G}_t \in \mathcal{X}, i_t \in \mathcal{Y}|h_t, a_t) = p(\tilde{G}_t \in \mathcal{X}|h_t, a_t)\, p(i_t \in \mathcal{Y}|h_t, a_t).$$

which establishes

$$\tilde{G}_t \perp\!\!\!\perp i_t | h_t, a_t.$$

Hence,

$$e_{\tilde{G}_t}^{\text{null}} \perp\!\!\!\perp i_t | h_t, a_t.$$

This concludes the proof. $\qquad\square$

Thus, the surrogate residual distribution

$$p(e_{\tilde{G}_t}^{\text{null}}, e_{i_t}, h_t, a_t) = p(h_t, a_t)\, p(e_{\tilde{G}_t}^{\text{null}}|h_t, a_t)\, p(e_{i_t}|h_t, a_t)$$

matches the ideal residual-based null in the conditional mean of the residual while enforcing conditional independence (CI).

Therefore, the permutation test operates under a first-moment-matching surrogate null rather than the ideal CI null. Nevertheless, the test targets violations of residual independence and allows for identifying privileged signals that provide additional predictive information about returns. Moreover, the proposed test has the following interpretation:

- If CI holds, then residuals are conditionally independent given $(h_t, a_t)$, and the test does not reject $\mathbb{H}_0^{\text{res}}$ asymptotically under standard regularity conditions.

- If the test rejects $\mathbb{H}_0^{\mathrm{res}}$, then $i_t$ contains predictive information about the conditional mean of $G_t$ beyond what is captured by $\mathbb{E}[G_t|h_t, a_t]$.

- The test does not guarantee full CI, only the absence of residual predictive information.

However, this notion of informativeness is in line with the objective of value-function learning, where conditional means, not full conditional distributions, determine optimal predictions.

### C.2. Effect of Regression Error on the Dependence Test

This subsection discusses how estimation errors in conditional expectations propagate to the population dependency measure underlying the residual-informativeness test.

**Regression error and residual perturbation.** Let $f_G(h_t, a_t) = \mathbb{E}[G_t|h_t, a_t]$ and $f_i(h_t, a_t) = \mathbb{E}[i_t|h_t, a_t]$ denote the true conditional expectations, and let $\hat{f}_G(h_t, a_t)$, $\hat{f}_i(h_t, a_t)$ be learned estimators. Define the regression errors

$$\Delta_{G_t} = f_G(h_t, a_t) - \hat{f}_G(h_t, a_t), \tag{11}$$

$$\Delta_{i_t} = f_i(h_t, a_t) - \hat{f}_i(h_t, a_t). \tag{12}$$

We assume the regression errors have finite second moments, i.e., $\mathbb{E}[\Delta_{G_t}^2] < \infty$, and $\mathbb{E}[\Delta_{i_t}^2] < \infty$.

The population residuals are

$$e_{G_t} = G_t - f_G(h_t, a_t), \qquad e_{i_t} = i_t - f_i(h_t, a_t),$$

while the estimated residuals are

$$\hat{e}_{G_t} = G_t - \hat{f}_G(h_t, a_t) = e_{G_t} + \Delta_{G_t}, \qquad \hat{e}_{i_t} = i_t - \hat{f}_i(h_t, a_t) = e_{i_t} + \Delta_{i_t}.$$

Throughout this section, we assume sample splitting, following the procedure described in Algorithm 1, so that regression estimation is independent of the dependence test.

Let $\rho(X, Y)$ denote a population dependency measure between square-integrable random variables. We impose the following stability condition.

**Assumption C.4** (Lipschitz continuity in $L^2$). There exists a constant $L_\rho > 0$ such that for all $X, X', Y, Y'$ with finite second moments,

$$\left|\rho(X, Y) - \rho(X', Y')\right| \leq L_\rho\left(\|X - X'\|_2 + \|Y - Y'\|_2\right),$$

where $\|Z\|_2 := \sqrt{\mathbb{E}[Z^2]}$.

This assumption is satisfied by many kernel- and distance-based dependency measures under additional regularity conditions (i.e., bounded Lipschitz kernels and finite second moments).

Applying Assumption C.4 with

$$(X, Y) = (e_{G_t}, e_{i_t}), \qquad (X', Y') = (\hat{e}_{G_t}, \hat{e}_{i_t}),$$

yields

$$\left|\rho(e_{G_t}, e_{i_t}) - \rho(\hat{e}_{G_t}, \hat{e}_{i_t})\right| \leq L_\rho\left(\|\Delta_{G_t}\|_2 + \|\Delta_{i_t}\|_2\right). \tag{13}$$

Thus, regression error perturbs the population dependency measure by at most an additive margin proportional to the $L^2$ regression errors.

## D. Environments

In this section, we detail the partially observable environments used in our experiments.

### D.1. Navigation Environments

This subsection presents the partially observable benchmark navigation environments used to assess the learning performance of the proposed informed asymmetric actor-critic method.

**Heaven-Hell-3.**  The Heaven-Hell task (Geffner & Bonet, 1998; Baisero, 2019) is a partially observable navigation problem in a grid-world environment with a corridor-like structure that forks into three distinct branches. Two of these branches correspond to terminal exits: one leading to a positive outcome (heaven) and the other to a negative outcome (hell). The third branch leads to a non-terminal location where the agent can interact with an oracle (priest) who provides information needed to disambiguate the exits. The agent is initially unaware of which terminal corresponds to heaven.

The underlying state includes both the agent's position and the true location of the heaven exit. As observation, however, the agent either perceives its own location or, when visiting the priest, receives an observation that reveals the location of heaven. We construct privileged partial information by adding to the agent's location its distance to the heaven terminal using Manhattan distance.

At each time step, the agent selects an action from the discrete set NORTH, SOUTH, EAST, WEST. The environment is deterministic, and movement is constrained by the grid-world layout. To solve the task optimally, the agent must first visit the priest to acquire the necessary information about the correct exit, then return to the fork and proceed to the identified heaven location.

The agent receives sparse feedback in the form of a terminal reward: a reward of 1.0 for exiting to heaven, and a reward of -1.0 for exiting to hell.

**Shopping-5.**  The Shopping-5 environment (Baisero, 2019) models another grid-world navigation task in which an agent must buy a forgotten item from a store. The environment is modeled as a two-dimensional grid-world of size $5 \times 5$, with the item placed randomly at one of the grid cells. The agent begins at an arbitrary location and must first locate and then buy the item. While the agent's position is fully observable, the item's position is hidden and must be explicitly queried.

Hence, the full state encodes both the agent's position and the item's location, represented compactly as integers. Observations are similarly encoded, but are partial: at each time step, the agent observes either its own position or, upon executing a query, the position of the item. Similar to the Heaven-Hell task, we introduce a privileged information by computing the current Manhattan distance between the agent and the item.

At each time step, the agent selects an action from the discrete set {UP, DOWN, LEFT, RIGHT, QUERY, BUY}. The four movement actions update the agent's position deterministically within the bounds of the grid. Executing the QUERY action returns the location of the item, but is subject to a cost. The BUY action attempts to purchase the item at the agent's current position; if executed in the correct cell, it completes the task successfully.

The environment provides a dense reward signal to encourage efficient behavior: a reward of -1.0 for moving, a reward of -2.0 for querying the item's location, a reward of -5.0 for a BUY action in the wrong cell, and a reward of +10.0 for a BUY action in the correct cell.

Optimal behavior requires the agent to query the item's location once, retain that information internally, and efficiently navigate to the target cell before executing a successful BUY action.

**Car-Flag.**  The Car-Flag environment (Nguyen, 2021) models a continuous control task where an agent controls a car moving along a one-dimensional track via discrete force-control actions. At the two ends of the track are terminal flags: one corresponding to a positive outcome (good flag) and the other to a negative outcome (bad flag). Reaching either flag terminates the episode. Additionally, an intermediate information flag is placed along the track; when reached, it reveals the position of the good flag. While the task is conceptually similar to Heaven-Hell, key differences are the force-control and the position of the information flag.

Both the state and observation spaces are represented as three-dimensional real-valued vectors. The state includes the agent's position, velocity, and the position of the good flag. The observation mirrors the state structure, but the third component (i.e., the good flag's position) is masked, i.e., set to zero, when the agent is outside the observation range of the information flag; and the agent's velocity is always hidden. In the informed setting, we provide the agent its velocity as a privileged partial signal.

At each time step, the agent selects an action from a discrete set of seven force-control inputs: LEFT_HIGH, LEFT_MEDIUM, LEFT_LOW, RIGHT_LOW, RIGHT_MEDIUM, RIGHT_HIGH, and NONE. These actions apply varying levels of acceleration to the left or right, or maintain zero acceleration.

The environment provides a sparse, terminal reward signal: a reward of 1.0 for reaching the good flag, and a reward of -1.0

for reaching the bad flag.

Optimal behavior requires the agent to first locate the information flag to identify the correct goal, then apply appropriate force controls to reach the good flag while avoiding the bad one.

**Cleaner.** Originally designed as a two-agent cooperative task, the Cleaner environment (Jiang & Amato, 2021) is adapted in this work to a single-agent control problem via fully centralized training and execution. In this formulation, the joint actions and observations are constructed via the Cartesian product of the corresponding spaces of the two individual agents. The environment is a maze-like $13 \times 13$ grid-world in which two robots must collectively cover and clean the entire area. The task is considered complete once every non-wall cell has been visited by at least one of the agents.

The full environment state is represented as a binary tensor of shape $13 \times 13 \times 5$, where each channel encodes the presence of: (i) a wall, (ii) a dirty cell, (iii) a cleaned cell, (iv) the first agent, and (v) the second agent. Each agent's local observation is a $3 \times 3 \times 3$ binary tensor that captures the immediate neighborhood centered around the agent, including information about walls, dirty cells, and clean cells. As privileged input, the critic receives a $13 \times 13 \times 5$ tensor encoding the agent's own position within the grid world, while masking out the position of the other agent by setting its corresponding cells to zero.

Each agent independently selects from four movement actions: UP, DOWN, LEFT, and RIGHT. In the centralized setting, where both agents are controlled jointly, the action space is the Cartesian product of the individual action sets, yielding a total of 16 composite actions.

At each time step, the agent receives a reward proportional to the number of new cells cleaned during that step. The possible reward values are: a reward of 0.0 if no new cells are cleaned, a reward of 1.0 if one agent cleaned a new cell, and a reward of 2.0 if both agents cleaned a new cell.

**Memory-Four-Rooms.** The so-called Gridverse suite (Baisero & Katt, 2021b) defines a collection of partially observable environments in which agents interact within structured grid-worlds. In this work, we consider the $7 \times 7$-Memory-Four-Room and $9 \times 9$-Memory-Four-Room environments. While actions are encoded as categorical indices, both states and observations are structured representations comprising multiple semantically meaningful components. Importantly, these components differ between state and observation, and some are only available in the state representation. The key components are:

- Grid component: A tensor of shape $3 \times 7 \times 7$ for $7 \times 7$-Memory-Four-Room or $3 \times 9 \times 9$ for $9 \times 9$-Memory-Four-Room, where each channel encodes a semantic property of the environment (e.g., cell type, cell color, or status). The observation includes a rotated, agent-centric $3 \times 2 \times 3$-view of this grid rendered from the first-person perspective of the agent. Cells obstructed by walls are occluded in the observation.

- Agent-ID-Grid component: A binary matrix of size $7 \times 7$ or $9 \times 9$, respectively, indicating the agent's absolute position. This component is included only in the state.

- Agent component: A three-dimensional categorical array encoding the agent's position and orientation. In the state, this is expressed in absolute coordinates, while in the observation, it is provided relative to the agent's perspective and is thus constant, and not necessary for control.

The environment contains a good exit, a bad exit, and a beacon, each placed randomly at the start of each episode. The beacon shares its color with the good exit, and successful task completion requires the agent to first locate the beacon, memorize its color, and then navigate to the exit of matching color while avoiding the bad exit.

As privileged input, the critic is provided with an agent-centered $3 \times 3 \times 5$ tensor, offering an expanded view of the agent's local surroundings.

At each time step, the agent selects from the following discrete action set: MOVE_FORWARD, MOVE_BACKWARD, MOVE_LEFT, MOVE_RIGHT, TURN_LEFT, TURN_RIGHT, PICK_N_DROP, and ACTUATE. The MOVE_ actions are interpreted relative to the agent's orientation, while TURN_ modifies the orientation itself. Although the action set includes PICK_N_DROP and ACTUATE for generality, these are no-ops in the Memory-Four-Rooms tasks, as there are no doors or pickable objects.

The reward signal is composed of the following terms: a living reward of -0.05 per time step, a reward of +5.0 for reaching the good exit, and a reward of -5.0 for reaching the bad exit.

### D.2. POPGym Environments

This subsection presents the partially observable environments from the POPGym suite (Morad et al., 2023) used to benchmark our informed asymmetric actor-critic method.

**Higher Lower.**   The Higher Lower task is a partially observable card game, in which the agent must predict whether the next drawn card has a higher or lower rank than the current card. At each step, the agent observes the current card and selects one of two discrete actions, HIGHER or LOWER, corresponding to the prediction that the next card will be of higher or lower rank, respectively.

After each action, a new card is drawn and revealed, serving as the reference card for the next decision. The process continues over a deck of cards, requiring the agent to remember previous card values to make correct decisions.

The reward is scaled by the deck size: correct predictions yield a positive reward, incorrect predictions a negative reward, and ties (identical ranks) result in zero reward.

We consider four variants of privileged information for this environment. The *full-state* signal provides the complete internal card-count representation of the environment state. The *expert* corresponds to an expert policy revealing whether the next card is higher or lower than the current card, effectively encoding the optimal prediction. The *previous-card* signal provides access to the previously observed card rank, while the *both-cards* signal additionally includes the current observed card together with the previous one.

**Repeat First.**   Repeat First is a partially observable card-game-based memory task. At the beginning of each episode, the agent observes an initial card suit that must be memorized throughout the episode. Subsequently, the agent receives a sequence of additional card suits and, at each step, must output the suit of the initial card.

The environment therefore requires the agent to retain information over long time horizons while processing distracting intermediate observations. At each time step, the agent selects one of four discrete actions corresponding to the card suits SPADES, HEARTS, DIAMONDS, and CLUBS.

The reward is scaled by the deck size: correctly predicting the initial card yields a positive reward, whereas incorrect predictions yield a negative reward.

We consider four variants of privileged information for this environment. The *hand* signal reveals the suit of the current card in hand. The *dealt* signal provides statistics over already dealt cards (i.e., frequency of observed suits). The *full-state* signal combines both sources of information. Finally, the *first-card* signal provides access to the suit of the first observed card in the sequence.

**Repeat Previous.**   Repeat Previous is a partially observable card-game-based memory task. The agent observes a sequence of card suits and, at each time step, must predict the suit observed $k$ steps earlier in the sequence. In our experiments, we use $k = 4$, requiring the agent to maintain a memory of past observations while processing continuously arriving inputs.

At each step, the agent selects one of four discrete actions corresponding to the four standard card suits: SPADES, HEARTS, DIAMONDS, and CLUBS. Rewards are scaled by the deck size: correctly recalling the target suit yields a positive reward, whereas incorrect predictions yield a negative reward.

We consider the same four variants of privileged information as in the Repeat First task, with the only difference that the *first-card* signal is replaced by the *previous-card* signal.

**Count Recall.**   Count Recall is a partially observable card-game-based memory task, designed to test order-agnostic memory and counting ability. At each time step, the agent observes a pair of cards corresponding to a currently drawn card and a query card. The task requires the agent to predict how many times the queried card has appeared in the history of previously observed cards.

At each step, the agent selects an integer-valued action corresponding to the predicted count. Rewards are scaled by the deck size: correct predictions of the queried count yield a positive reward, while incorrect predictions yield a negative reward.

We consider six variants of privileged information for this environment. The *all-values* signal provides access to the underlying identities of all cards in the deck. The *is-face-up* signal indicates which cards are currently revealed. The

*first-card* and *second-card* signals provide the identities of the most recently selected cards. The *flipped-cards* signal provides both of the last two selected cards jointly. Finally, the *full-state* signal combines card identities, visibility information, and the most recent selection history.

**Concentration.**    Concentration is a partially observable card-matching task, inspired by the classic memory game of the same name. A full deck of cards is initially placed face-down, and the agent interacts with the environment by selecting cards to flip.

At each time step, the agent chooses a card to reveal. If two consecutively selected cards match (according to the chosen matching criterion, e.g., in rank or color), the pair is removed from the game and the agent receives a positive reward. Otherwise, the selection is penalized and the cards are flipped back face-down.

We consider four variants of privileged information for this environment. The *all-values* signal provides full access to the underlying card identities. The *is-face-up* signal indicates which cards are currently revealed. The *first-card* and *second-card* signals provide access to the most recently selected cards.

**Position Cart Pole.**    The position-only Cart Pole task is a partially observable variant of the classic control benchmark (Barto et al., 1983). The environment implements the standard Cart Pole mechanics, but the agent does not observe cart position and pole angle velocities directly. Instead, it receives only the position and pole angle, requiring it to infer missing dynamical information over time.

At each time step, the agent selects a discrete action corresponding to the standard Cart Pole control inputs: PUSH-LEFT or PUSH-RIGHT. Successful balancing yields a small positive reward scaled by the episode length, while failure results in a negative reward.

We consider four variants of privileged information for this environment. The *x-velocity* signal provides access to the cart's horizontal velocity. The *angle-velocity* signal provides access to the angular velocity of the pole. The *both-velocities* signal provides both velocity components jointly. Finally, the *full-state* signal reveals the full environment state.

### D.3. Synthetic Informed POMDPs

We generate a distribution of synthetic informed POMDP instances with a finite state space of size $|\mathcal{S}|$, a discrete action space of size $|\mathcal{A}|$, and continuous observation and information spaces. Following the methodology of François-Lavet et al. (2019), transition probabilities are randomly assigned by setting each $(s_t, a_t, s_{t+1})$-entry to zero with probability $0.75$, and sampling uniformly from $[0, 1]$ otherwise. To ensure valid transitions, we assign a non-zero probability to a randomly chosen next state whenever all transitions from a given state-action pair are initially zero. We then normalize the probabilities so that they sum to one. Each state is associated with a Gaussian feature vector $s_t \in \mathbb{R}^{d_s}$ with $d_s \in \mathbb{Z}_{>0}$: $s_t \sim \mathcal{N}(\mathbf{0}, \sigma_s^2 \mathbf{I}_{d_s})$, where $\mathbf{0}$ is the $d_s$-dimensional zero vector, $\mathbf{I}_{d_s}$ the $d_s \times d_s$ identity matrix, and $\sigma_s^2$ the variance. Rewards are linear functions of the state features, with reward weights $w_r \in \mathbb{R}^{d_s}$ sampled uniformly from $[-1, 1]$ at initialization.

We first construct the privileged information $i_t \in \mathbb{R}^{d_i}$ by selecting a subset of $1 \leq d_i \leq d_s$ state features using a binary masking vector $x_i \in \{0, 1\}^{d_s}$ and applying a selection matrix $W_i \in \{0, 1\}^{d_i \times d_s}$:

$$i_t = W_i(x_i \odot s_t)$$

where $\odot$ denotes element-wise multiplication. Observations $o_t \in \mathbb{R}^{d_o}$ are generated by masking a subset of $1 \leq d_o \leq d_i$ features from $i_t$ using a binary masking vector $x_o \in \{0, 1\}^{d_i}$, applying a selection matrix $W_o \in \{0, 1\}^{d_o \times d_i}$ to the masked privileged signal, and adding Gaussian noise:

$$o_t = W_o(x_o \odot i_t) + \beta_o \epsilon_o, \qquad \epsilon_o \sim \mathcal{N}(\mathbf{0}, \sigma_o^2 \mathbf{I}_{d_o}),$$

where $\beta_o \geq 0$ modulates the observation noise and $\sigma_o^2$ is the variance.

## E. Model Architectures and Hyperparameters

In this section, we describe the actor and critic architectures as well as the hyperparameters used in our experiments.

### E.1. Navigation Tasks

We use the implementation ((Baisero & Katt, 2021a; Baisero, 2019; Baisero & Katt, 2021b; Nguyen, 2021) of environments and actor-critic methods provided by Baisero & Amato (2022), extending them to the informed setting.

In each task, a 128-dimensional single-layer gated recurrent unit (GRU) encodes the concatenated action and observation features into a history representation. While the actor and critic networks share this architectural component, their parameters are maintained separately. The subsequent actor and critic network components differ across environments as follows:

- For the Heaven-Hell-3 and Shopping-5 tasks, we employ a 64-dimensional embedding model to represent states, actions, and observations. Both the actor and critic networks consist of two-layer feedforward neural networks with 512 and 256 units, respectively, using ReLU activations in the hidden layers and a linear output layer.

- For the Car-Flag and Cleaner environments, actions are represented as one-hot encodings of their respective categorical indices. As the state and observation representations provided by these environments are already flattened and structurally simple, no additional embedding is applied. The actor's and critic's subsequent networks adopt the same architecture used for the Heaven-Hell-3 and Shopping-5 tasks.

- For the Memory-Four-Room tasks, the $3 \times 2 \times 3$ observation tensors are initially processed by an embedding layer that maps each categorical value to an 8-dimensional vector. The resulting embedded tensor is then flattened into a 144-dimensional feature vector, which serves as the observation input to both the actor and critic networks. Actions, provided as categorical indices, are represented using one-dimensional embedding layers. For the states, the grid component is first embedded and then concatenated with the agent-ID grid. A three-layer convolutional network subsequently processes this combined input. The output of the convolutional network is concatenated with the agent components. The actor and critic networks each consist of a hidden layer with 512 units using ReLU activation, followed by a linear output layer.

We encode the privileged information analogously to the observations. The embedded privileged information is then concatenated with the latent history representation before being passed to the task-specific feedforward neural network.

*Table 2.* Hyperparameters for the benchmark navigation environments.

| ENVIRONMENT | $\eta_\pi$ | $\eta_{\hat{V}}$ | $\lambda_0$ |
|---|---|---|---|
| HEAVEN-HELL-3 | 0.001 | 0.001 | 0.1 |
| SHOPPING-5 | 0.001 | 0.0003 | 3.0 |
| CAR-FLAG | 0.001 | 0.001 | 0.03 |
| CLEANER | 0.001 | 0.001 | 1.0 |
| $7 \times 7$-MEMORY-FOUR-ROOM | 0.0003 | 0.001 | 0.1 |
| $9 \times 9$-MEMORY-FOUR-ROOM | 0.001 | 0.0003 | 0.3 |

For each environment and method, we use the hyperparameter values recommended by Baisero & Amato (2022) to ensure comparability with prior work. Table 2 summarizes the actor learning rate $\eta_\pi$, critic learning rate $\eta_{\hat{V}}$, and the initial negative-entropy weight $\lambda_0$ selected for each environment. Additionally, the following model hyperparameters are applied across all environments: discount factor is set to $\gamma = 0.99$, episodes are automatically terminated if they exceed 100 time steps; two episodes are sampled per gradient update; a frozen target network is used to stabilize critic training, with target parameters updated every 10,000 time steps; and the negative-entropy weight $\lambda$ decays linearly over 2 million time steps to a final value equal to one-tenth of $\lambda_0$.

### E.2. POPGym Environments

We use the environment implementations of Morad et al. (2023), and adopt their recommended actor and critic architectures, extending them to the informed setting.

For each environment and method, a 256-dimensional single-layer GRU encodes the interaction history into a fixed-length representation. Before being processed by the GRU, observations are projected into a 128-dimensional zero-mean, unit-variance representation using a linear layer followed by layer normalization and a LeakyReLU activation. The GRU hidden

state is projected onto a 128-dimensional feature vector, which is fed into separate actor and critic heads. Both heads are implemented as two-layer feedforward networks with 128 hidden units and LeakyReLU activations. For the A2C variants with an informed or full-state critic, we encode the privileged signal analogously to the observations. The resulting embedding is concatenated with the history representation before being passed to the critic head.

We use the hyperparameter values recommended by Morad et al. (2023) for the actor-critic architecture to ensure comparability with prior work. We set the learning rate to $\eta = 5 \times 10^{-4}$, use backpropagation-through-time truncation length of 1,024, a zero entropy regularization weight $\lambda = 0$, and a discount factor of $\gamma = 0.99$. For the encoding of the privileged signal, we use an embedding size of 64 for the card-game tasks and an embedding size of 128 for the position-only Cart Pole environment.

### E.3. Synthetic Informed POMDPs

For the synthetic informed POMDP environments, the actor is implemented as a single-layer GRU with hidden size 64, followed by a linear readout layer. Both the symmetric and informed history critics use separate single-layer GRU with hidden size 64 to produce fixed-length representations of the interaction history, followed by a linear readout layer that outputs the value estimate. For the informed history critic, the GRU hidden state is concatenated with the privileged signal before being passed to the linear layer. The learning rate for both the actor and the critics is set to $\eta = 1 \times 10^{-4}$, and we use a fixed discount factor of $\gamma = 0.99$ across all environments.

## F. Wall-Clock Runtimes

All experiments were conducted on a cluster node equipped with 96 cores running at 3.0 GHz and 93.75 GB of RAM allocated per task. Tables 3 and 4 report the wall-clock training times for different A2C variants and privileged signals across the navigation and PopGym environments, respectively. Table 5 summarizes the wall-clock runtimes for the experiments in Section 5.2.

*Table 3.* Wall-clock training time across the six benchmark navigation tasks for different A2C variants. We report absolute training time (in seconds) and relative runtime w.r.t. the symmetric A2C (100%). Means and standard deviations are computed across 20 independent runs.

| ENVIRONMENT | ALGORITHM | WALL-CLOCK TIME (s) | RELATIVE TIME (%) |
|---|---|---|---|
| MEMORY-FOUR-ROOMS-7X7 | A2C | 2.18e+04 ± 1.52e+03 | 100.00 |
| | INFORMED-ASYM-A2C | 2.32e+04 ± 1.66e+03 | 106.04 |
| | ASYM-A2C-S | 2.88e+04 ± 2.29e+03 | 132.03 |
| | ASYM-A2C-HS | 3.45e+04 ± 2.28e+03 | 158.02 |
| MEMORY-FOUR-ROOMS-9X9 | A2C | 2.13e+04 ± 1.70e+03 | 100.00 |
| | INFORMED-ASYM-A2C | 2.38e+04 ± 1.19e+03 | 111.62 |
| | ASYM-A2C-S | 2.76e+04 ± 1.54e+03 | 129.48 |
| | ASYM-A2C-HS | 3.53e+04 ± 1.10e+03 | 165.93 |
| HEAVEN-HELL-3 | ASYM-A2C-S | 5.81e+03 ± 1.23e+03 | 72.75 |
| | A2C | 7.99e+03 ± 1.47e+03 | 100.00 |
| | INFORMED-ASYM-A2C | 8.97e+03 ± 1.66e+03 | 112.27 |
| | ASYM-A2C-HS | 9.15e+03 ± 1.34e+03 | 114.45 |
| SHOPPING-5 | ASYM-A2C-S | 2.16e+03 ± 2.94e+02 | 56.71 |
| | A2C | 3.81e+03 ± 7.73e+02 | 100.00 |
| | ASYM-A2C-HS | 4.28e+03 ± 5.81e+02 | 112.27 |
| | INFORMED-ASYM-A2C | 4.35e+03 ± 3.08e+02 | 113.96 |
| CAR-FLAG | ASYM-A2C-S | 4.38e+03 ± 1.08e+03 | 72.26 |
| | INFORMED-ASYM-A2C | 5.94e+03 ± 1.44e+03 | 97.93 |
| | A2C | 6.07e+03 ± 1.67e+03 | 100.00 |
| | ASYM-A2C-HS | 6.20e+03 ± 1.74e+03 | 102.19 |
| CLEANER | ASYM-A2C-S | 5.39e+03 ± 3.42e+02 | 90.66 |
| | A2C | 5.95e+03 ± 4.46e+02 | 100.00 |
| | ASYM-A2C-HS | 6.20e+03 ± 2.46e+02 | 104.26 |
| | INFORMED-ASYM-A2C | 6.23e+03 ± 3.29e+02 | 104.68 |

*Table 4.* Wall-clock training time across the six POPGym environments for different privileged signals. We report absolute training time (in seconds) and relative runtime w.r.t. the symmetric A2C (100%). Means and standard deviations are computed across 20 independent runs.

| ENVIRONMENT | PRIVILEGED SIGNAL | WALL-CLOCK TIME (s) | RELATIVE TIME (%) |
|---|---|---|---|
| | NONE | 8.76e+03 $\pm$ 1.36e+02 | 100.00 |
| | FULL STATE | 9.46e+03 $\pm$ 1.04e+02 | 107.93 |
| HIGHER LOWER | PREVIOUS-CARD | 9.51e+03 $\pm$ 7.47e+01 | 108.51 |
| | BOTH-CARDS | 9.52e+03 $\pm$ 6.90e+01 | 108.66 |
| | EXPERT | 9.53e+03 $\pm$ 7.82e+01 | 108.80 |
| | NONE | 9.12e+03 $\pm$ 8.92e+02 | 100.00 |
| | HAND | 9.66e+03 $\pm$ 1.00e+03 | 105.90 |
| REPEAT PREVIOUS | DEALT | 9.76e+03 $\pm$ 1.04e+03 | 106.92 |
| | FULL STATE | 9.91e+03 $\pm$ 8.90e+02 | 108.62 |
| | PREVIOUS-CARD | 9.97e+03 $\pm$ 8.70e+02 | 109.30 |
| | NONE | 9.14e+03 $\pm$ 7.97e+02 | 100.00 |
| | DEALT | 9.70e+03 $\pm$ 9.52e+02 | 106.10 |
| REPEAT FIRST | HAND | 9.76e+03 $\pm$ 8.91e+02 | 106.85 |
| | FULL STATE | 9.80e+03 $\pm$ 9.87e+02 | 107.19 |
| | FIRST-CARD | 9.95e+03 $\pm$ 9.31e+02 | 108.86 |
| | NONE | 1.18e+04 $\pm$ 7.38e+02 | 100.00 |
| | EXPERT | 1.22e+04 $\pm$ 4.24e+02 | 103.88 |
| COUNT RECALL | FREQ-CARDS-SEEN | 1.23e+04 $\pm$ 4.95e+02 | 104.79 |
| | FREQ-CARDS-QUERIED | 1.24e+04 $\pm$ 4.88e+02 | 105.12 |
| | FULL STATE | 1.24e+04 $\pm$ 4.61e+02 | 105.26 |
| | LAST-OBS | 1.26e+04 $\pm$ 4.53e+02 | 107.47 |
| | NONE | 6.67e+03 $\pm$ 4.80e+02 | 100.00 |
| | FLIPPED-CARDS | 6.90e+03 $\pm$ 4.40e+02 | 103.48 |
| | SECOND-CARD | 6.97e+03 $\pm$ 3.95e+02 | 104.50 |
| CONCENTRATION | FIRST-CARD | 7.04e+03 $\pm$ 4.43e+02 | 105.46 |
| | ALL-VALUES | 7.15e+03 $\pm$ 4.49e+02 | 107.17 |
| | FULL STATE | 7.17e+03 $\pm$ 3.58e+02 | 107.44 |
| | IS-FACE-UP | 7.25e+03 $\pm$ 2.94e+02 | 108.73 |
| | NONE | 8.87e+03 $\pm$ 7.90e+01 | 100.00 |
| | FULL STATE | 9.06e+03 $\pm$ 1.02e+02 | 102.14 |
| POSITION CART POLE | BOTH-VELOCITIES | 1.01e+04 $\pm$ 2.51e+02 | 113.34 |
| | X-VELOCITY | 1.02e+04 $\pm$ 6.57e+01 | 115.48 |
| | ANGLE-VELOCITY | 1.10e+04 $\pm$ 1.83e+01 | 123.62 |

*Table 5.* Wall-clock runtime comparison of the two informativeness tests across different privileged signals $i_t$. Means and standard deviations are computed across 10 independent runs.

| PRIVILEGED SIGNAL $i_t$ | WALL-CLOCK RUNTIME (s) | |
|---|---|---|
| | Residual Informativeness Test | Prediction Informativeness Test |
| $i_t = [s_t^1, s_t^2]$ | 939.70 $\pm$ 43.76 | 898.24 $\pm$ 65.57 |
| $i_t = [s_t^1, s_t^2, s_t^3]$ | 936.96 $\pm$ 42.71 | 906.89 $\pm$ 56.61 |
| $i_t = [s_t^1, s_t^2, s_t^4]$ | 964.76 $\pm$ 39.17 | 910.51 $\pm$ 48.68 |
| $i_t = [s_t^1, s_t^2, s_t^5]$ | 950.20 $\pm$ 31.40 | 913.10 $\pm$ 69.74 |
| $i_t = [s_t^1, s_t^2, s_t^3, s_t^4]$ | 939.95 $\pm$ 46.23 | 890.51 $\pm$ 75.44 |
| $i_t = [s_t^1, s_t^2, s_t^3, s_t^5]$ | 931.06 $\pm$ 30.45 | 889.61 $\pm$ 82.82 |
| $i_t = [s_t^1, s_t^2, s_t^4, s_t^5]$ | 919.11 $\pm$ 50.37 | 905.27 $\pm$ 55.26 |
| $i_t = [s_t^1, s_t^2, s_t^3, s_t^4, s_t^5]$ | 946.33 $\pm$ 21.69 | 896.10 $\pm$ 48.37 |

# G. Additional Experiments

This section summarizes our additional empirical results.

## G.1. Learning Performance on POPGym Environments

Table 6 reports the final performance (mean episodic return) and area under the learning curve (AUC) after 2 million steps for actor-critic variants with access to different privileged signals on the POPGym environments.

*Table 6.* Learning performance of actor-critic variants with different privileged signals on the POPGym environments, computed as mean ± standard deviation over 20 independent runs.

| ENVIRONMENT | PRIVILEGED SIGNAL | FINAL PERFORMANCE | AUC |
|---|---|---|---|
| HIGHER LOWER | FULL STATE | 4.90e-01 ± 6.91e-02 | 9.59e+05 ± 1.99e+05 |
| | EXPERT | 4.92e-01 ± 9.53e-02 | 9.59e+05 ± 1.99e+05 |
| | BOTH-CARDS | 4.81e-01 ± 9.23e-02 | 9.58e+05 ± 1.97e+05 |
| | PREVIOUS-CARD | 5.02e-01 ± 1.20e-01 | 9.55e+05 ± 1.98e+05 |
| | NONE | 4.83e-01 ± 7.82e-02 | 9.38e+05 ± 1.26e+05 |
| REPEAT PREVIOUS | PREVIOUS-CARD | 3.79e-01 ± 3.02e-01 | 6.81e+05 ± 5.40e+05 |
| | DEALT | 4.77e-01 ± 7.46e-02 | 8.28e+05 ± 2.88e+05 |
| | NONE | 4.60e-01 ± 1.20e-01 | 8.36e+05 ± 2.64e+05 |
| | FULL STATE | 5.06e-01 ± 7.44e-02 | 8.75e+05 ± 2.58e+05 |
| | HAND | 5.02e-01 ± 3.70e-02 | 9.14e+05 ± 1.91e+05 |
| REPEAT FIRST | NONE | 8.29e-01 ± 5.34e-01 | 6.63e+05 ± 1.18e+06 |
| | HAND | 5.51e-01 ± 8.15e-01 | 5.38e+05 ± 1.34e+06 |
| | FIRST-CARD | 3.45e-01 ± 9.20e-01 | 2.86e+05 ± 1.44e+06 |
| | DEALT | 8.76e-01 ± 4.52e-01 | 1.66e+05 ± 1.41e+06 |
| | FULL STATE | 9.00e-01 ± 4.47e-01 | 1.96e+04 ± 1.47e+06 |
| COUNT RECALL | FREQ-CARDS-QUERIED | 7.18e-01 ± 1.41e-01 | 1.57e+06 ± 2.19e+05 |
| | LAST-OBS | 7.75e-01 ± 1.19e-01 | 1.61e+06 ± 2.27e+05 |
| | FULL STATE | 7.53e-01 ± 1.10e-01 | 1.63e+06 ± 1.93e+05 |
| | FREQ-CARDS-SEEN | 7.69e-01 ± 1.32e-01 | 1.66e+06 ± 2.07e+05 |
| | NONE | 8.45e-01 ± 1.13e-01 | 1.76e+06 ± 1.68e+05 |
| | EXPERT | 8.84e-01 ± 1.01e-01 | 1.82e+06 ± 1.29e+05 |
| CONCENTRATION | ALL-VALUES | 1.45e-01 ± 7.59e-02 | 3.49e+05 ± 2.13e+05 |
| | SECOND-CARD | 1.57e-01 ± 1.11e-01 | 3.79e+05 ± 2.24e+05 |
| | FIRST-CARD | 1.29e-01 ± 9.48e-02 | 3.89e+05 ± 2.43e+05 |
| | NONE | 2.38e-01 ± 1.39e-01 | 4.22e+05 ± 2.32e+05 |
| | IS-FACE-UP | 2.20e-01 ± 1.06e-01 | 4.71e+05 ± 2.73e+05 |
| | FLIPPED-CARDS | 2.59e-01 ± 2.79e-01 | 4.91e+05 ± 4.69e+05 |
| | FULL STATE | 4.24e-01 ± 4.03e-01 | 7.00e+05 ± 5.60e+05 |
| POSITION CART POLE | ANGLE-VELOCITY | 5.82e-01 ± 2.95e-01 | 8.68e+05 ± 4.64e+05 |
| | BOTH-VELOCITIES | 5.37e-01 ± 3.45e-01 | 8.48e+05 ± 4.78e+05 |
| | FULL STATE | 5.69e-01 ± 2.61e-01 | 8.26e+05 ± 4.20e+05 |
| | X-VELOCITY | 1.65e-01 ± 1.05e-01 | 2.57e+05 ± 1.49e+05 |
| | NONE | 5.55e-02 ± 2.09e-02 | 1.53e+05 ± 7.50e+04 |

## G.2. Effect of Privileged Signal Choice on Policy Performance

We study how different privileged signal generators affect the quality of the learned policy. To this end, we train a symmetric actor-critic baseline and multiple informed asymmetric actor-critic (IAAC) agents in the same fixed randomly sampled environment, each IAAC variant using a different privileged signal as described in Section 5.2. Each model is trained for 15,000 gradient steps, where every step uses a batch of 16 episodes. Every 50 gradient steps, we evaluate the current policy by estimating the mean episodic return over 50 evaluation episodes. All experiments are repeated with 10 random seeds. Figure 2 presents the actor loss, critic loss and episodic return on evaluation episodes during training for selected actor-critic variants, smoothed using a moving average over 500 episodes.

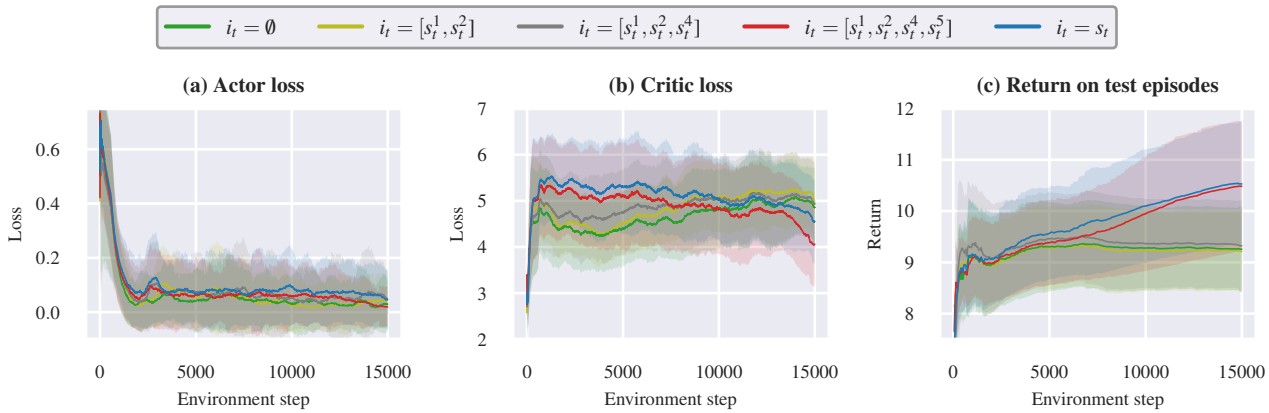

*Figure 2.* Learning performance of selected actor-critic variants on a randomly sampled informed POMDP. The curves depict (a) the actor loss, (b) the critic loss, and (c) episodic returns evaluated on 50 test episodes. The results are smoothed using a moving average over 500 environment steps, with the means and standard deviations computed across 10 independent runs.

For each privileged signal generator, we compute the average gain in mean episodic return relative to the symmetric critic across all evaluation episodes during training, averaged over the 10 runs. In Figure 3, we plot this performance gain against the corresponding effect sizes of the two informativeness measures (observed HSIC and value prediction gain) to illustrate the relationship between signal informativeness and control performance. The informativeness tests are conducted offline using separate episodes collected with a random policy, allowing us to assess how well the proposed criteria serve as a proxy for downstream policy improvement.

The upper-right quadrant of the scatter plots highlights the most relevant privileged signal generators, combining high estimated informativeness with strong gains in mean episodic return. Signals such as $i_t = [s_t^1, s_t^2, s_t^4, s_t^5]$ and $i_t = s_t$ are consistently identified as informative (low $p$-values: 0.05-residual informativeness and $(0, 0.05)$- or $(0, 0.1)$-prediction informativeness) and yield the largest performance improvements. In contrast, some signals (e.g., $i_t = [s_t^1, s_t^2, s_t^3, s_t^4]$ or $i_t = [s_t^1, s_t^2, s_t^4]$) show strong effect sizes but smaller gains in episodic returns, while $i_t = [s_t^1, s_t^2, s_t^3, s_t^5]$ provides higher episodic gains despite lower measured informativeness. Signals identical to the observations ($i_t = o_t = [s_t^1, s_t^2]$) remain uninformative, in line with their limited contribution to policy improvement.

These results show that, while the proposed informativeness criteria are designed to quantify the predictive value of privileged signals for return estimation, they also serve as a reasonable proxy for downstream policy improvement. Notably, the tests are performed on episodes collected with a random policy, yet signals identified as informative tend to yield higher gains in episodic return during policy training. This suggests that value-based informativeness captures essential aspects of signals that facilitate control, while also highlighting the need for future research to develop criteria that directly assess a signal's potential to enhance policy performance beyond value prediction. Interestingly, the full-state signal $i_t = s_t$ does not always yield the highest episodic return gains, further highlighting the practical relevance of the informed asymmetric actor-critic framework, which can exploit any state-dependent privileged information beyond full-state access.

### G.3. Noisy observations

In addition to the noiseless observation setting studied in Section 5.2, we assess our informativeness criteria under noisy observation configurations. Recall that we generate observations by masking a subset of features from $i_t$ and adding Gaussian noise modulated by $\beta_o \geq 0$.

Tables 7 and 8 report results for multiple privileged signal generators under noise levels $\beta_o = 0.1$ and $\beta_o = 0.5$, averaged over ten independent runs, using the $\alpha$-residual-informativeness criterion and the post-hoc $(0, \delta)$-prediction-informativeness measure, respectively. Increasing observation noise from $\beta_o = 0.0$ to $\beta_o = 0.1$ or $\beta_o = 0.5$ has only a minor impact on the relative ranking of privileged signals under both informativeness criteria. Signals that include the most return-relevant state components (notably those containing $s_t^4$) remain consistently identified as informative, with low $p$-values in both the residual-based and prediction tests. In contrast, signals lacking these components (e.g., $[s_t^1, s_t^2]$) continue to appear uninformative across noise levels. While effect sizes tend to increase slightly with higher noise for the truly informative

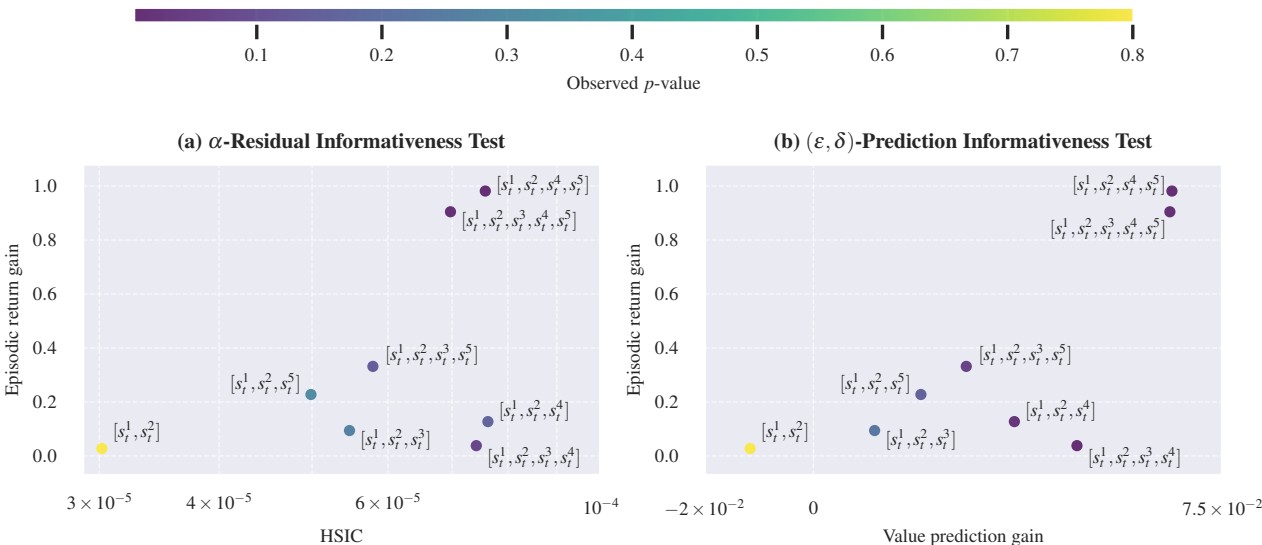

*Figure 3.* Relationship between estimated signal informativeness and downstream control performance. Each point corresponds to a distinct privileged signal configuration. The x-axis shows the effect size of the (a) residual-based informativeness measure (observed HSIC) and (b) post-hoc prediction informativeness measure (value prediction gain), while the y-axis shows the average gain in mean episodic return relative to a symmetric actor-critic baseline, aggregated over all evaluation episodes and averaged across 10 runs. Color of the point indicates the corresponding $p$-value of the respective informativeness test.

signals the statistical conclusions remain generally unchanged.

This trend is intuitive. As observation noise increases, the history $h_t$ becomes a less accurate proxy for the latent state, making the history-conditioned belief over states more uncertain. Privileged signals that provide additional state-conditioned information can therefore explain a larger fraction of the remaining return variance, increasing their relative informativeness compared to the history alone.

An exception is $i_t = [s_t^1, s_t^2, s_t^3]$ under the residual-based criterion: its mean $p$-value increases noticeably with higher noise, making it less likely to be deemed as informative. In contrast, the post-hoc prediction criterion shows, on average, an increase in effect size for this signal at higher noise, indicating that it still provides useful predictive information even when the residual test becomes more conservative.

## G.4. Alternative feature subset

To assess the robustness of our results with respect to the choice of feature subsets used as observations, we repeat the informativeness tests from Section 5.2 using an alternative feature subset, with $\beta_o = 0.0$, while keeping the reward weights unchanged. This change in configuration also induces different signal generators, as we employ privileged signals of the form $i_t = (o_t, o_t^+)$. Comparing results across feature subsets allows us to evaluate the extent to which our findings depend on the particular choice of observable features.

Table 9 summarizes the performance of different privileged signal generators under both informativeness criteria, averaged over ten independent runs. For this analysis, the agent is provided with observations $o_t = [s_t^1, s_t^3]$. In line with the previous experiments, signals that add little beyond the observation (e.g., $i_t = [s_t^1, s_t^3]$) are identified as uninformative by both criteria, with near-zero effect sizes and large $p$-values. Incorporating return-relevant components increases effect sizes. Under the post-hoc prediction criterion, signals containing strongly reward-relevant features (i.e., $s_t^4$ and combinations including $s_t^2$ and $s_t^4$) yield clear and statistically significant gains, reflected in larger positive $L_\tau$ and small $p$-values. This supports earlier findings that signals which include the most reward-informative state dimensions are consistently ranked highest. The residual-based criterion remains more conservative, as in the original observation setting. Although the dependency measure $\rho_{\text{obs}}$ generally increases when informative features are added, statistical significance at level $\alpha = 0.1$ is typically reached only for larger feature subsets (e.g., those containing both $s_t^2$ and $s_t^4$, or the full state $s_t$).

Overall, the relative ordering of privileged signal generators is largely preserved across observation choices: signals containing the most reward-relevant state components remain the most informative under both criteria, although in some cases, higher significance levels are required to reach statistical significance.

*Table 7.* Comparison of candidate privileged signals using the residual-based informativeness criterion across ten independent runs on a randomly sampled informed POMDP environment with noisy observations.

| PRIVILEGED SIGNAL $i_t$ | $\beta_o = 0.1$ | | $\beta_o = 0.5$ | |
|---|---|---|---|---|
| | $\rho_{\text{obs}}$ (mean $\pm$ std) | $p$-value (mean $\pm$ std) | $\rho_{\text{obs}}$ (mean $\pm$ std) | $p$-value (mean $\pm$ std) |
| $i_t = [s_t^1, s_t^2]$ | 3.2e-05 $\pm$ 1.2e-05 | 0.406 $\pm$ 0.280 | 3.8e-05 $\pm$ 1.8e-05 | 0.380 $\pm$ 0.321 |
| $i_t = [s_t^1, s_t^2, s_t^3]$ | 8.4e-05 $\pm$ 3.5e-05 | 0.069 $\pm$ 0.120 | 8.7e-05 $\pm$ 3.9e-05 | 0.105 $\pm$ 0.223 |
| $i_t = [s_t^1, s_t^2, s_t^4]$ | 1.2e-04 $\pm$ 3.7e-05 | 0.011 $\pm$ 0.003 | 1.3e-04 $\pm$ 3.8e-05 | 0.010 $\pm$ 0.000 |
| $i_t = [s_t^1, s_t^2, s_t^5]$ | 4.2e-05 $\pm$ 1.5e-05 | 0.262 $\pm$ 0.303 | 7.0e-05 $\pm$ 4.0e-05 | 0.088 $\pm$ 0.087 |
| $i_t = [s_t^1, s_t^2, s_t^3, s_t^4]$ | 1.1e-04 $\pm$ 3.7e-05 | 0.018 $\pm$ 0.017 | 1.2e-04 $\pm$ 3.7e-05 | 0.012 $\pm$ 0.006 |
| $i_t = [s_t^1, s_t^2, s_t^3, s_t^5]$ | 7.2e-05 $\pm$ 2.7e-05 | 0.061 $\pm$ 0.065 | 8.2e-05 $\pm$ 3.5e-05 | 0.043 $\pm$ 0.062 |
| $i_t = [s_t^1, s_t^2, s_t^4, s_t^5]$ | 9.8e-05 $\pm$ 2.8e-05 | 0.015 $\pm$ 0.013 | 1.2e-04 $\pm$ 4.0e-05 | 0.012 $\pm$ 0.006 |
| $i_t = [s_t^1, s_t^2, s_t^3, s_t^4, s_t^5]$ | 9.5e-05 $\pm$ 3.0e-05 | 0.013 $\pm$ 0.009 | 1.1e-04 $\pm$ 3.5e-05 | 0.013 $\pm$ 0.007 |

*Table 8.* Comparison of candidate privileged signals using the post-hoc prediction informativeness criterion across ten independent runs on a randomly sampled informed POMDP environment with noisy observations.

| PRIVILEGED SIGNAL $i_t$ | $\beta_o = 0.1$ | | $\beta_o = 0.5$ | |
|---|---|---|---|---|
| | $L_\tau$ (mean $\pm$ std) | $p$-value (mean $\pm$ std) | $L_\tau$ (mean $\pm$ std) | $p$-value (mean $\pm$ std) |
| $i_t = [s_t^1, s_t^2]$ | -0.013 $\pm$ 0.014 | 0.788 $\pm$ 0.343 | 0.004 $\pm$ 0.012 | 0.456 $\pm$ 0.298 |
| $i_t = [s_t^1, s_t^2, s_t^3]$ | 0.007 $\pm$ 0.016 | 0.384 $\pm$ 0.330 | 0.021 $\pm$ 0.014 | 0.081 $\pm$ 0.084 |
| $i_t = [s_t^1, s_t^2, s_t^4]$ | 0.035 $\pm$ 0.018 | 0.041 $\pm$ 0.071 | 0.049 $\pm$ 0.017 | 0.001 $\pm$ 0.002 |
| $i_t = [s_t^1, s_t^2, s_t^5]$ | 0.019 $\pm$ 0.019 | 0.219 $\pm$ 0.231 | 0.036 $\pm$ 0.017 | 0.030 $\pm$ 0.031 |
| $i_t = [s_t^1, s_t^2, s_t^3, s_t^4]$ | 0.046 $\pm$ 0.018 | 0.009 $\pm$ 0.016 | 0.057 $\pm$ 0.017 | 3.2e-04 $\pm$ 5.9e-04 |
| $i_t = [s_t^1, s_t^2, s_t^3, s_t^5]$ | 0.027 $\pm$ 0.019 | 0.110 $\pm$ 0.137 | 0.043 $\pm$ 0.017 | 0.010 $\pm$ 0.011 |
| $i_t = [s_t^1, s_t^2, s_t^4, s_t^5]$ | 0.064 $\pm$ 0.020 | 0.003 $\pm$ 0.006 | 0.077 $\pm$ 0.018 | 1.5e-04 $\pm$ 3.3e-04 |
| $i_t = [s_t^1, s_t^2, s_t^3, s_t^4, s_t^5]$ | 0.056 $\pm$ 0.032 | 0.073 $\pm$ 0.131 | 0.068 $\pm$ 0.030 | 0.022 $\pm$ 0.041 |

*Table 9.* Comparison of candidate privileged signals using residual-based and post-hoc prediction informativeness criteria across ten independent runs with observations $o_t = [s_t^1, s_t^3]$.

| PRIVILEGED SIGNAL $i_t$ | RESIDUAL INFORMATIVENESS | | PREDICTION INFORMATIVENESS | |
|---|---|---|---|---|
| | $\rho_{\text{obs}}$ (mean $\pm$ std) | $p$-value (mean $\pm$ std) | $L_\tau$ (mean $\pm$ std) | $p$-value (mean $\pm$ std) |
| $i_t = [s_t^1, s_t^3]$ | 2.4e-05 $\pm$ 6.6e-06 | 0.673 $\pm$ 0.273 | -1.9e-02 $\pm$ 0.010 | 0.973 $\pm$ 0.041 |
| $i_t = [s_t^1, s_t^2, s_t^3]$ | 4.9e-05 $\pm$ 1.8e-05 | 0.240 $\pm$ 0.315 | 0.023 $\pm$ 0.010 | 0.052 $\pm$ 0.099 |
| $i_t = [s_t^1, s_t^3, s_t^4]$ | 4.5e-05 $\pm$ 1.6e-05 | 0.314 $\pm$ 0.329 | 0.008 $\pm$ 0.015 | 0.307 $\pm$ 0.345 |
| $i_t = [s_t^1, s_t^3, s_t^5]$ | 5.8e-05 $\pm$ 4.7e-05 | 0.307 $\pm$ 0.275 | 0.004 $\pm$ 0.014 | 0.390 $\pm$ 0.342 |
| $i_t = [s_t^1, s_t^2, s_t^3, s_t^4]$ | 6.2e-05 $\pm$ 1.6e-05 | 0.057 $\pm$ 0.082 | 0.057 $\pm$ 0.012 | 0.003 $\pm$ 0.008 |
| $i_t = [s_t^1, s_t^2, s_t^3, s_t^5]$ | 5.4e-05 $\pm$ 3.3e-05 | 0.240 $\pm$ 0.279 | 0.039 $\pm$ 0.012 | 0.016 $\pm$ 0.031 |
| $i_t = [s_t^1, s_t^3, s_t^4, s_t^5]$ | 6.8e-05 $\pm$ 3.8e-05 | 0.080 $\pm$ 0.076 | 0.031 $\pm$ 0.018 | 0.107 $\pm$ 0.192 |
| $i_t = [s_t^1, s_t^2, s_t^3, s_t^4, s_t^5]$ | 6.3e-05 $\pm$ 3.0e-05 | 0.071 $\pm$ 0.102 | 0.068 $\pm$ 0.019 | 0.006 $\pm$ 0.013 |

