# OpenReview forum: "Informed Asymmetric Actor-Critic: Leveraging Privileged Signals Beyond Full-State Access"
_ICML.cc/2026/Conference — ICML 2026 regular_

### Official Review · Reviewer_Frrd · 2026-03-08

**Soundness:** 3
**Presentation:** 3
**Significance:** 3
**Originality:** 3
**Overall Recommendation:** 4
**Confidence:** 4

**Summary:**

The paper introduces an Informed Asymmetric Actor-Critic (IAAC) framework in which the actor remains conditioned only on observable histories, while the critic can use arbitrary state-dependent privileged signals during training, rather than requiring the full state. The central theoretical claim is that this broader class of privileged signals still yields an unbiased policy gradient, extending prior asymmetric actor-critic formulations beyond full-state critics.

Beyond the generalized framework, the paper asks which privileged signals are actually useful, and proposes two criteria for evaluating signal quality: a pre-training residual informativeness test based on dependence between return residuals and privileged-signal residuals, and a post-hoc prediction informativeness criterion based on improvement in value prediction accuracy. These are intended to provide a principled way to compare candidate privileged signals from the perspective of critic learning and return prediction.

Empirically, the paper evaluates IAAC on six benchmark POMDP navigation tasks against symmetric and asymmetric A2C baselines.

**Compliance With Llm Reviewing Policy:**

Affirmed.

**Key Questions For Authors:**

1. To what extent do your proposed informativeness criteria predict downstream policy improvement, rather than only improved value prediction?
2. How robust is the proposed framework to privileged signals that are only weakly informative, redundant, or partially misleading?

**Limitations:**

yes

**Strengths And Weaknesses:**

Strengths:
- The paper addresses a meaningful and well-motivated problem in partially observable reinforcement learning: how to exploit privileged training-time information in asymmetric actor-critic methods without requiring access to the full latent state. This is a real conceptual gap in prior asymmetric actor-critic formulations, which often assume full-state access or otherwise rely on stronger assumptions about the relationship between history and state.
- The paper does not stop at the “unbiasedness” result, but asks the more practically relevant question of which privileged signals are worth using. The two proposed informativeness criteria—one based on residual dependence and one based on return prediction gain—are sensible attempts to operationalize signal quality. In particular, the residual-based criterion is attractive because it can be evaluated prior to RL training under an arbitrary exploratory policy, which gives the paper some methodological value beyond the formal theorem alone.

Weaknesses:
- The privileged signals used in the benchmark tasks are hand-designed and task-specific, such as Manhattan distance to the target, velocity, or enlarged local observations. These are reasonable design choices for demonstrating the idea, but they also make the setting somewhat curated. The paper would be more compelling if it provided stronger evidence that the proposed criteria can discover or validate useful privileged signals in less structured settings, rather than mainly confirming intuitions in carefully chosen ones.

---

> ### Author Rebuttal · Authors · 2026-03-30
>
> We thank the reviewer for their valuable feedback and for recognizing our contribution in extending the admissible
> class of privileged signals to any state-conditioned information. We agree that selecting a relevant signal is
> critical, and our proposed criteria address this challenge. Furthermore, the ability to leverage
> non-Markovian signals offers a significant practical advantage. Below, we address the reviewer's concerns and questions.
>
> **W1: Hand-crafted signals & generalizability.** We agree that validation in less structured settings is essential.
> While our benchmark tasks (Sec. 5.1) use manually designed signals for clarity, which may appear curated, this mirrors
> standard practice in RL, where designers explicitly define observation and action spaces. Our framework simply extends
> this design choice to include remaining training-time variables (or subsets thereof) as privileged signals. Just as
> observation design is task-specific, so is privileged signal design.
>
> Crucially, our criteria remove the need for perfect human intuition for signal selection. We dedicate Sec. 5.2 and
> Appendix F to validating that our criteria can discover useful signals in less structured settings, such as synthetic
> informed POMDPs. In these environments, reward-relevant features
> were stochastically generated and unknown a priori. Our criteria successfully identified relevant subsets (e.g., $s_t^4$
> and $s_t^5$)
> automatically from dozens of candidates in a data-driven way. Additionally, our recent experiments on the POPGym
> benchmark [1] confirm that privileged information is beneficial in more complex environments where human intuition
> may be misleading when selecting privileged signals (see also our response to Reviewer z3XN, W3).
> **We propose to expand the discussion in Sec. 5.2 to explicitly highlight this ability to discover useful signals
> without prior human curation.**
>
> **Q1: Do criteria predict downstream policy improvement?** Yes. While formally defined for value prediction, our
> criteria serve as a proxy for downstream policy improvement. As shown in Appendix F.1 (Fig. 3), signals with high
> informativeness scores consistently yield the largest gains in episodic return relative to symmetric baselines. **To
> make this explicit in the camera-ready version, we propose to add a "regret" column to Tab. 1 (e.g., reporting the area
> under the learning curve), directly linking informativeness scores to learning performance.**
>
> **Q2: Robustness to weak, redundant, or misleading signals.** Our framework ensures robustness via two mechanisms.
> - _Theoretical convergence_: Per Thm. 3.1, the informed policy gradient is unbiased for any state-conditioned signal.
>   Even with a weak signal, both symmetric and informed methods optimize the identical objective $J(\pi_\theta)$ and
>   asymptotically converge to the same stationary points $\theta^\*$ where $\nabla_\theta J(\pi_\theta)=0$.
>   This aligns with recent theoretical insights regarding asymmetric settings [2]. Signal choice does not
>   compromise the asymptotic optimality of the policy; the worst-case outcome is simply a lack of variance reduction
>   effect and less sample-efficient learning, not divergence per se. **We propose to highlight this in Sec. 3.**
> - _Empirical filtering_: Our criteria prevent the use of suboptimal signals by discarding redundant or uninformative
>   signals. Notably, we found that carefully selected subsets can outperform the full state, as our
>   criteria discarded noisy features that hinder critic generalization. Thus, our criteria effectively filter out poor
>   signals. In our experiments in synthetic informed POMDPs (Sec. 5.2), signals lacking return-relevant components or
>   containing redundant signals (e.g., $i_t=o_t$) consistently yield larger $p$-values and near-zero effect sizes. The results in Appendix F.1 confirm that such signals provide negligible policy improvement over the symmetric baseline, indicating that our criteria can distinguish between useful and redundant/misleading information.
>   Crucially, we added a robustness ablation: we introduced pure-Gaussian-noise privileged signals of different
>   dimensionality. The residual-based test yielded mean $p$-values of $0.28 \pm 0.19$ for such a 2-dimensional signal
>   and $0.43 \pm 0.29$ for a 5-dimensional signal, respectively; well above any significance threshold.
>   Furthermore, our additional experiments using the POPGym environments support the observation that redundant
>   information does not accelerate learning.
>   **We propose to include these findings and a summary of the additional empirical results in Sec. 5.**
>
> We hope that these discussions will have answered your questions. Please do not hesitate to ask us to elaborate further if anything remains unclear.
>
> - [1] https://arxiv.org/abs/2303.01859
> - [2] https://arxiv.org/abs/2501.19116

---

> > ### Author Rebuttal · Reviewer_Frrd · 2026-04-01
> >
> > I am very grateful to the authors for answering my questions.

---

> > > ### Author Response · Authors · 2026-04-03
> > >
> > > We thank the reviewer once again for their valuable feedback, for the careful reading of our response, and for
> > > confirming that all raised concerns and questions have been adequately addressed.
> > >
> > > If there are any remaining concerns that prevented a score increase to 5 ("accept"), please do not hesitate to mention
> > > them so that we can provide additional clarification or make minor revisions to further strengthen the work.

---

### Official Review · Reviewer_J9Fu · 2026-03-10

**Soundness:** 3
**Presentation:** 3
**Significance:** 2
**Originality:** 3
**Overall Recommendation:** 4
**Confidence:** 3

**Summary:**

This paper is theoretically grounded and proposes a generalization of asymmetric actor-critic for POMDPs. The paper theoretically shows that any privileged signal is acceptable, and proposes two criteria for deciding which signals are useful. However, it might be hard to find useful signals, and in the experimental environments, the signals are manually chosen.

**Compliance With Llm Reviewing Policy:**

Affirmed.

**Final Justification:**

I maintain a Weak Accept recommendation. The rebuttal was helpful and addressed several of my questions, including clarification of when privileged signals may improve learning, discussion of sensitivity and implementation details, correction of a technical typo, and additional runtime reporting planned for the final version.

**Key Questions For Authors:**

1. It is better to end-to-end signal selection on the tasks, rather than manully choose the privileged signal manually and  validate only in synthetic POMDPs?
2. Could you clarify that when the privileged signals improve sample efficiency or variance in function approximation?
3. Need the sensitivity analysis of the residual informativeness test to the history encoder and data coverage.
4. Line 189, the inequality writing seems to uses variance of the state not the $G_t$, which might confuse readers.
5. How sensitive are the gains to the chosen privileged signal, LR pair, entropy coefficient?
6. Could the authors report wall-clock training time?

**Limitations:**

Yes.

**Strengths And Weaknesses:**

**Strengths**
1. The theorem is useful. It removes the need for full-state access and thus supports the claim of arbitrary state-dependent training-time signals.

**Weaknesses**
1. The privileged signals are manually designed for each environment, rather than being chosen end-to-end automatically.
2. A hard issue is finding useful and accessible privileged signals. “Diagnostic sensors, internal simulator variables, or forecasted data” also seem unrealistic in the real world.

---

> ### Author Rebuttal · Authors · 2026-03-30
>
> We thank the reviewer for their valuable feedback and for recognizing the value of Thm. 3.1 in relaxing the requirement of full-state access. Below, we address the reviewer's concerns and questions.
>
> **Admissible privileged signals.** We proved that any state-conditioned information is admissible; this class is
> broad, but not completely arbitrary or unrestricted (e.g., information about future states or actions, available in
> hindsight, provides biased gradients without further assumptions).
> Regarding history-dependent signals $i^h_t\sim p(\cdot|h_t)$, our framework could indeed be generalized to those. Their
> utility might be limited as they do not add information about the state, and therefore offer only limited
> variance reduction. **We will clarify this in the camera‑ready version.**
>
> **W1 & W2: Hand-crafted signals & accessibility.** Regarding hand-crafted signals, please refer to our response to
> Reviewer Frrd, W1. Regarding availability, privileged signals are increasingly available via multi-view sensors in
> robotics [1], via access to simulator states (available for any training in simulation), or even via external knowledge
> provided by foundation models (e.g., querying large-model representations based on the observations and eventual
> additional context). Our theory provides a principled way to leverage these complex, state-conditioned signals without
> requiring them to be Markovian. **We consider that there exist many types of additional information that may be
> leveraged, and propose to expand the discussion on future work in Sec. 1 or 6.**
>
> **Q1: End-to-end selection.** While end-to-end optimization is interesting, our work lays the foundations by
> decoupling signal selection from policy optimization. Embedding selection within the RL loop comes with additional
> non-stationarity and the risk of unstable and less efficient learning. We demonstrated that our residual-based criterion
> can rank candidate signals using (few) data from a (cheap) arbitrary policy (Sec. 4 & 5.2), effectively automating
> selection before a more costly training. **However, we acknowledge end-to-end methods as future work and will add a
> short discussion in Sec. 6.**
>
> **Q2: When does a signal reduce variance/regret?** Thm. 3.1 guarantees unbiased gradients for any state‑conditioned
> signal, but faster convergence requires variance reduction. Conditioning the critic on $i_t$ reduces TD-target variance
> by the law of total variance: $E[Var(G_t|h_t,a_t,i_t)]\leq E[Var(G_t|h_t,a_t)]$.
> It mostly happens in environments with high "value aliasing", where distinct states yielding different returns may
> exist under the same observable history [2]. In such cases, $i_t$ can help to disambiguate the state, providing a
> cleaner target for the critic.
> However, this benefit appears only if $i_t$ is genuinely informative about the return, and its processing cost does not
> outweigh the variance gain. **We will expand the discussion in Sec. 3 to link variance reduction to the expected
> speed-up in convergence.**
>
> **Q3: Sensitivity analysis of test.** Our residual-based test relies on estimating conditional expectations over the
> full history. We compress each history into a fixed‑length vector with an RNN, but any encoder (fixed‑window,
> padding, etc.) would work. A practical benefit of the RNN is that it can be reused during training for the
> critic networks.
>
> As detailed in Appendix C.2, for Lipschitz-continuous dependence measures, we can bound the
> perturbation of residuals based on the regression error. If the regression estimators are not consistent (i.e., errors
> do not vanish as sample size increases), the test may suffer from inflated false positive rates or reduced power. To
> mitigate this, we employ cross-fitting and use Random Forests, which are less prone to overfitting than
> deep networks with limited data.
>
> **Q4: Typo in line 189.** We thank the reviewer for catching this error. The inequality should use $G_t$ instead
> of $s_t$. **We will correct this.**
>
> **Q5: Sensitivity to signals & hyperparameters.**
>
> - _Signal choice_: Gains are highly sensitive to signal content. As shown in Tab. 1, privileged signals that contain
>   return‑relevant features ($s_t^4$, $s_t^5$) yield significant prediction gains; irrelevant ones do not. This
>   underscores the necessity of our selection criteria.
> - _Hyperparameters_: For the benchmark tasks, we use the same tuned hyperparameters as the symmetric baseline [3].
>   Importantly, our method does not add any additional hyperparameters.
>
> **Q6: Wall-clock training times.** We agree that runtime is an important metric. **We will include a new table in the
> camera-ready manuscript reporting the average wall-clock training times for all methods across benchmarks.**
>
> We hope these answers fully settle all open questions. Please do not hesitate to request additional explanations.
>
> - [1] https://arxiv.org/abs/1710.06542
> - [2] https://arxiv.org/abs/2501.19116
> - [3] https://arxiv.org/abs/2105.11674

---

> > ### Author Rebuttal · Reviewer_J9Fu · 2026-04-02
> >
> > Thank you for the detailed rebuttal. Manually designed privileged signals and the practical accessibility are still a concern. While you provide plausible sources, the current experiments still rely on hand-crafted signals, and it remains unclear how your framework would perform when such signals are not readily available or must be learned automatically.

---

> > > ### Author Response · Authors · 2026-04-03
> > >
> > > We thank the reviewer once again for their valuable feedback, and for the follow-up questions regarding manually
> > > designed signals and the practical accessibility. Below, we answer these questions.
> > >
> > > **Hand-crafted signals and subsets of state features.** In the benchmark navigation tasks (Sec. 5.1) and the POPGym
> > > environments [1], the simulator provides the ground-truth state $s_t$. The privileged partial signals used in our
> > > experiments are obtained via a mapping $i_t=f(s_t)$, which extracts a low-dimensional representation of the state.
> > > These signals are computed in the simulator, but unavailable during execution (e.g., Manhattan distance to the target,
> > > or a cropped view of the agent's surroundings). Since this mapping depends solely on the hidden state $s_t$, the policy
> > > gradients remain unbiased (cf. Thm. 3.1).
> > > The term "hand-crafted" refers here only to the choice of the low-dimensional feature fed to the critic, not
> > > external data acquisition. For instance, our residual-based criterion ranks for `Shopping-5` the Manhattan
> > > distance-based signals with an effect size of $0.0013\pm0.0003$ ($p$-value: $0.0099\pm0.0$) slightly higher than
> > > full-state signals ($0.0004\pm0.0001$; $p$-value: $0.0109\pm0.0031$).
> > >
> > > Furthermore, as highlighted in our synthetic informed POMDP experiments (Sec. 5.2) and additional POPGym results,
> > > selecting meaningful (i.e., return-relevant) subsets of state features is also a valid strategy when full-state access
> > > is available in simulation.
> > >
> > > **Automated signal selection.**
> > > To automate the design of $f$, we can enumerate various functions of $s_t$, rank them using our informativeness
> > > criteria,
> > > and pick the top-ranked signal(s). Thus, the user only provides a set of potential signal generators; the
> > > algorithm automatically discovers which ones seem beneficial.
> > >
> > > **Handling limited, high-dimensional, or absent privileged sources.** The reviewer raises an important
> > > question regarding how our method handles situations where manually designed candidates are unavailable, leaving us with
> > > only limited or a single high-dimensional source of additional information (e.g., high-dimensional simulator state,
> > > multi-view sensor data, or large-model embeddings).
> > > Notably, Thm 3.1 does not make any assumptions about the dimensionality or Markovian structure of the privileged
> > > state-conditioned signal; an aspect that, to our knowledge, has not been previously addressed in the literature. Hence,
> > > the resulting informed asymmetric actor-critic still yields unbiased gradient estimates.
> > > However, obtaining meaningful privileged information in such settings often requires learning a compact
> > > embedding $g(x_t)$ of the raw privileged data $x_t$ on a modest offline buffer (e.g., using a shallow MLP
> > > or an auto-encoder). This learned embedding is then fed to the critic, just as with ready-to-use signals.
> > > While this "learn-then-use"step does not affect the correctness of the algorithm, it introduces an additional
> > > representation-learning stage that we consider beyond the scope of this paper.
> > >
> > > In summary, a practical approach would involve (1) collecting a modest offline buffer from a "cheap" exploratory
> > > policy, (2) learning different low-dimensional embeddings, (3) ranking the signal generators using our proposed
> > > criteria, and (4) training the informed actor-critic with the top-ranked generator(s).
> > >
> > > When neither additional sensors nor simulator state are accessible during training, the asymmetric learning setting
> > > might not apply, simply collapsing to the standard symmetric actor-critic. Nevertheless, our criteria can still help
> > > identify whether feeding $i_t=o_t$ (or a subset thereof) to the critic, in addition to the learned history
> > > representation, is beneficial for the critic learning process.
> > >
> > > **We propose to add these explanations to Sec. 4 in the camera-ready manuscript.**
> > >
> > > **Our contribution and future work.** We acknowledge that manually designing privileged signals can be impractical for a
> > > single high-dimensional additional information source, or immediate selection might be impossible without learning a
> > > meaningful representation. Therefore, "learning" a useful signal first seems like a sensible approach, which
> > > we consider a promising direction for future work. However, in this work, we have already established a solid foundation
> > > by:
> > >
> > > - Proving that any state-conditioned additional information $i_t$ yields unbiased policy gradients (not just the state $s_t$),
> > > - Demonstrating empirically that certain privileged partial signals $i_t$ can outperform the full state $s_t$,
> > > - Providing criteria for identifying such useful $i_t$,
> > > - Validating empirically that these criteria succeed in identifying these $i_t$.
> > >
> > > While automation and end-to-end selection are essential extensions that can be addressed in future work, we remain
> > > confident that our research has set the groundwork for advancements in this area.
> > >
> > > We hope that these clarifications will have answered your remaining concerns.
> > >
> > > - [1] https://arxiv.org/abs/2303.01859

---

### Official Review · Reviewer_z3XN · 2026-03-13

**Soundness:** 3
**Presentation:** 3
**Significance:** 3
**Originality:** 3
**Overall Recommendation:** 4
**Confidence:** 3

**Summary:**

This paper explore a core challenge in partially observable reinforcement learning: how to leverage privileged training-time information without assuming full-state access. This paper proposes an **Informed Asymmetric Actor-Critic** framework, introduces two criteria for evaluating the informativeness of privileged signals, and shows empirically that carefully chosen signals can match or outperform full-state asymmetric critics using less information

**Compliance With Llm Reviewing Policy:**

Affirmed.

**Key Questions For Authors:**

Q1: Can the authors provide more intuition or analysis on when a privileged signal is expected to improve learning speed versus merely preserving unbiasedness?

Q2:  What is the practical computational overhead of these signal-selection procedures compared to simply training the asymmetric critic directly?

Q3: Can the authors clarify when less information is actually better than full-state access, and whether this is mainly due to representation simplicity, reduced noise, or easier value fitting?

**Strengths And Weaknesses:**

**Strengths**
* The paper’s main theoretical contribution is clean and interesting: it shows that any state-conditioned privileged signal can be used by the critic without biasing the policy gradient, which meaningfully generalizes prior asymmetric actor-critic work beyond full-state critics.
* The paper goes beyond theory by asking a practical question: which privileged signals are actually useful? The proposed residual-informativeness and prediction-gain criteria are a nice step toward principled signal selection.
* The experiments are reasonably convincing: on several benchmark POMDP tasks, informed-asym-A2C often matches or outperforms asymmetric baselines, suggesting that well-chosen partial privileged signals can be as useful as full-state access.


**Weaknesses**
* While the theory guarantees unbiasedness, it does not fully explain when privileged signals lead to substantial variance reduction or better optimization in deep RL practice.

* The informativeness criteria are interesting, but they introduce extra machinery such as RNN encoders, cross-fitting, random forests, HSIC tests, and bootstrap/t-tests, which may make the framework somewhat heavy in practice.

* The experiments are promising, but still somewhat limited in scale, mostly focusing on benchmark navigation tasks and synthetic informed POMDPs.

* The paper argues that partial privileged signals can outperform full-state access in some cases, but this claim could use deeper explanation.

---

> ### Author Rebuttal · Authors · 2026-03-30
>
> We thank the reviewer for their valuable feedback and their accurate summary of our core contributions: the
> theoretical generalization to arbitrary state-conditioned signals and the practical utility of our proposed
> informativeness criteria. We are happy to read that the reviewer recognized that our experiments support our claims.
> Below, we address the reviewer's concerns and questions.
>
> **W1 & Q1: When do privileged signals improve learning?** Please refer to our response to Reviewer J9Fu, Q2.
>
> **W2 & Q2: Computational overhead of signal selection.** While we agree that the selection machinery introduces
> overhead, it is crucial to distinguish between the offline selection phase and the online training loop.
>
> - _Offline selection_: This one-time, pre-training step involving our residual-based informativeness test might be
>   costly initially, but its cost is amortized over the entire RL training run. If the selected signal reduces the sample
>   complexity, the upfront selection cost might become negligible compared to saved environment interactions and gradient
>   steps.
> - _Online training_: Once a signal is selected, the IAAC training loop concatenates the fixed-length signal (or its
>   embedding) to the history encoding, leading to minimal overhead compared to asymmetric critics.
> - _Scalability_: For large candidate sets or high-dimensional signals, the HSIC estimation can be accelerated using for
>   example the Nyström method [1].
>
> **We propose to add a discussion in Sec. 4 characterizing this trade-off and providing wall-clock times for the
> selection phase.**
>
> **W3: Limited scale of experiments.** We agree that our primary benchmarks mainly focus on navigation tasks and
> synthetic POMDPs. However, we have conducted additional experiments on the POPGym benchmark suite [2] to address
> scalability concerns, confirming that the IAAC maintains its performance advantages in more complex settings. **We will
> include a summary of these POPGym results in the camera-ready manuscript.**
>
> The table below summarizes some results of our additional experiments that we conducted during rebuttal phase using
> the POPGym benchmark suite. We report the mean $\pm$ standard deviation of the area under the learning curve (i.e., the sum
> of normalized mean episodic return across 5 independent runs, each involving 3,000,000 environment steps). Due to space
> constraints, we included only the best-performing informed critic per task in the table below (we evaluated several different
> privileged signals for each environment).
>
> |                  | `HigherLower`                   | `CountRecall`                     | `Concentration`                  | `PositionOnlyCartPole`          |
> |:-----------------|:--------------------------------|:----------------------------------|:---------------------------------|:--------------------------------|
> | Symmetric critic | $4.23 \pm 0.36$ ($\times 10^5$) | $-12.10 \pm 0.35$ ($\times 10^5$) | $-5.81 \pm 2.21$ ($\times 10^5$) | $0.79 \times 10^5 \pm 11.45$    |
> | Informed critic  | $4.26 \pm 0.36$ ($\times 10^5$) | $-11.76 \pm 0.43$ ($\times 10^5$) | $-3.29 \pm 1.04$ ($\times 10^5$) | $2.43 \times 10^5 \pm 230.84$ * |
>
> _*) The best-performing informed critic in `PositionOnlyCartPole` was using $i_t = s_t$. The second best-performing
> critic was only conditioned on a subset of state-features and achieved mean AUC of $2.08 \times 10^5 \pm 289.92$._
>
> **W4 & Q3: Why is less information sometimes better than full-state access?** Our empirical results show that carefully
> selected feature subsets of the state can outperform critics with full-state access. This is due to two factors:
>
> - _Noise filtering_: The full state may contain irrelevant features that need to be learned to ignore, increasing sample
>   complexity and the risk of overfitting. A selected signal improves signal-to-noise ratio, providing a cleaner,
>   low-variance target. For instance, in a "noisy TV" scenario where the full state includes a rapidly changing but
>   reward-irrelevant channel, a critic conditioned on the full state must waste capacity modeling this noise, whereas a
>   critic conditioned on a filtered signal ignores it entirely.
> - _Representation efficiency_: Simplifying the function approximation problem by selecting a sufficient statistic (as
>   guided by our criteria), thus reducing the input-space dimensionality, leads to faster convergence and better
>   generalization.
>
> **We will expand the discussion in Sec. 5 to highlight these mechanisms using our synthetic POMDP results and concrete
> examples.**
>
> We hope these answers fully settle all open questions. Please do not hesitate to ask us to elaborate further if anything remains unclear.
>
> - [1] https://arxiv.org/abs/2302.09930
> - [2] https://arxiv.org/abs/2303.01859

---

> > ### Author Rebuttal · Reviewer_z3XN · 2026-04-01
> >
> > Thanks for your response and but I will keep my rating 4.

---

> > > ### Author Response · Authors · 2026-04-03
> > >
> > > We thank the reviewer once again for their valuable feedback, for the careful reading of our response, and for
> > > confirming that all raised concerns and questions have been adequately addressed. We also appreciate that, after the
> > > clarifications, the reviewer has chosen to maintain their score of 4.
> > >
> > > If there are any remaining concerns that prevented a score increase to 5 ("accept"), please do not hesitate to mention
> > > them so that we can provide additional clarification or make minor revisions to further strengthen the work.

---

### Official Review · Reviewer_pLmq · 2026-03-16

**Soundness:** 3
**Presentation:** 3
**Significance:** 2
**Originality:** 2
**Overall Recommendation:** 4
**Confidence:** 4

**Summary:**

The paper proposes an asymmetric actor-critic algorithm that uses arbitrary state-conditioned information rather than full states in the previous asymmetric actor-critic literature. Two different selection criteria and accompanied algorithms are proposed: one on directly testing for residual independence to evaluate the informativeness of the selected information variable and another one on testing how much action-value prediction error could be reduced. Experiment results on several toy POMDP environments validates the learning efficiency of the proposed informed asymmetric actor-critic algorithm.

**Compliance With Llm Reviewing Policy:**

Affirmed.

**Final Justification:**

The rebuttal cleared my concerns. I have raised to 4.

**Key Questions For Authors:**

1. Could the authors provide additional experiments on more complex POMDPs like pixel based manipulation tasks discussed in [OGBench](https://seohong.me/projects/ogbench/)?
2. Will access to expert or better than random trajectories provide better information selection?
3. What is the relationship between those state factors in the synthetic informed POMDP used in sec 5.2?
4. There is a line of work from the causality literature discussing which subset of state factors are most relevant to the decision and should be observed for optimal performance, e.g., [Characterizing Optimal Mixed Policies: Where to Intervene and What to Observe](https://proceedings.neurips.cc/paper_files/paper/2020/file/61a10e6abb1149ad9d08f303267f9bc4-Paper.pdf). Could the authors also relate to those?
5. Lastly and most importantly, could the authors provide more justification in selecting partial state information instead of using full state in asymmetric actor-critic? The benefits are not clear from the current experiment results.

**Limitations:**

No.

**Strengths And Weaknesses:**

**Strength**
1. The perspective on using only partial state information in asymmetric actor-critic algorithm opens new discussion on input selection for better sample efficiency.
2. The presentation is easy to follow and has sufficient theoretical groundings.
3. For each information selection algorithm discussed, practical implementation discussion is provided alongside the formal definitions.

**Weakness**
1. Given the dominating trend of scaling, selecting only relevant inputs to improve learning efficiency seems to be less concerned nowadays. Some discussion and justification could provide stronger motivation for the work.
2. From my understanding of the contributions, only using partial information instead of full state access should mainly improve sample efficiency. However, the experiments use relatively simple environments and small neural networks making it hard to manifest the major advantage. This is also confirmed by the results in fig1. The blue line using full state/history is actually better than or on par with the proposed method in 4/6 environments. Only in one environment (Car-Flag), clear gap is observed.

---

> ### Author Rebuttal · Authors · 2026-03-30
>
> We thank the reviewer for their valuable feedback. We appreciated the recognition of the presentation clarity and the
> usefulness of our selection algorithms. Below, we address the concerns and questions.
>
> **W1 & Q5: Motivation for partial information.** We agree that feeding all available information to a large critic model
> is appealing. However, this approach has relied on two unrealistic assumptions before our work:
> - _Markovianity_: There is no guarantee that all additional information collectively forms a Markovian state, as
>   required by previous asymmetric actor-critic approaches. Our framework relaxes this requirement by allowing the critic
>   to use any state-conditioned information.
> - _Sample efficiency & noise filtering_: Even with large model capacity, training on irrelevant features (e.g.,
>   distractors, see our discussion about causality below) increases statistical complexity, slowing convergence and
>   increasing gradient variance. As shown in our synthetic experiments (Sec. 5.2), irrelevant state components can dilute
>   the reward-relevant features. Please also refer to our response to Reviewer z3XN, W4 & Q3.
>
> Selecting informative signals is crucial when computational budgets are limited or the full state is inaccessible. This
> is precisely where our selection criteria provide value. **We propose to add this discussion to Sec. 1.**
>
> **W2 & Q1: Experimental scale & performance gaps.** The goal of our experiments was threefold: to show that (1) learning
> with partial signals can surpass symmetric A2C performance, (2) it can sometimes surpass A2C with state access in terms
> of convergence speed, and (3) that our criteria can identify these scenarios.
> - _Convergence speed_: In `Shopping-5` and `Car-Flag`, using carefully selected privileged signals converges faster than
>   with full state or than the symmetric baseline. This supports the noise-filtering argument: as the optimal policies
>   are theoretically identical across these methods, our method converges faster by reducing the variance of TD-targets.
> - _Complexity_: Beyond standard grid-world tasks [1], we tried the POPGym suite [2] to address the scalability concerns.
>   The results confirm the sample efficiency advantages in more complex and memory-intensive settings (see also our
>   response to Reviewer z3XN, W3). **We will add these POPGym results in the camera-ready manuscript.**
>
> _OGBench_: We thank the reviewer for this suggestion. Unfortunately, OGBench targets offline RL, whereas our method
> focuses on online interaction; we thus preferred POPGym.
>
> **Q2: Expert policies.** Our residual-based method works with data from any exploratory policy (even random), enabling
> offline selection before costly training begins. While access to expert trajectories might improve the precision
> of the estimates by covering high-reward regions, they are not strictly necessary (see also our response to Reviewer
> Frrd, Q1).
>
> We thank the reviewer for their suggestion of considering expert policies, which significantly strengthens our
> theoretical argument: our framework even supports a state-conditioned expert policy as a privileged signal,
> i.e., $i_t=a^\*_t\sim\pi^\*(\cdot|s_t)$, akin to concepts in asymmetric imitation learning (but without bias!) [3].
> Since $a^\*_t$ is state-conditioned, Thm. 3.1 guarantees unbiased policy gradients for the history-dependent actor, even
> if the expert policy is not deployable due to partial observability. It allows the critic to leverage an oracle to
> reduce variance without direct imitation, which is known to be suboptimal [3]. **We will expand Sec. 3 to discuss this
> connection and include experiments in POPGym (e.g., `HigherLower`) using expert actions as privileged signals.**
>
> **Q3: Relationship between state factors.** In the synthetic POMDPs, the state consists of random Gaussian
> features, $s_t^k$. Rewards are linear functions of the state. In the presented instance, $s_t^4$ and $s_t^5$ were
> highly relevant; others were not. Our criteria successfully identified signals containing $s_t^4$ and $s_t^5$ as the most
> informative ones, confirming that reward-relevant components aid critic learning.
>
> **Q4: Relation to causality research.** We thank the reviewer for highlighting the connection to causality literature.
> Our work aligns with causality literature by suggesting that observing all context can be suboptimal if some do not (
> causally) influence rewards. We reach this conclusion statistically: conditioning the critic on the full state can
> introduce noise if irrelevant features are present, increasing value target variance. **We will discuss relevant
> causality works in Sec. 1 to position our contribution within this literature.**
>
> We hope that these discussions and the additional experiments that we plan to include will have addressed your concerns.
> Please do not hesitate to ask for further explanations.
>
> - [1] https://arxiv.org/abs/2105.11674
> - [2] https://arxiv.org/abs/2303.01859
> - [3] https://arxiv.org/abs/2012.15566

---

> > ### Author Rebuttal · Reviewer_pLmq · 2026-03-31
> >
> > Thanks for your clarifications. I'll raise to 4.

---

> > > ### Author Response · Authors · 2026-04-03
> > >
> > > We thank the reviewer once again for their valuable feedback, for the careful reading of our response, and for
> > > confirming that all raised concerns and questions have been adequately addressed. We are pleased that, after the
> > > clarifications, the reviewer has raised their score to 4.
> > >
> > > If there are any remaining concerns that prevented a score increase to 5 ("accept"), please do not hesitate to mention them so that we can provide additional clarification or make minor revisions to further strengthen the work.

---

### Decision · Program_Chairs · 2026-04-30

**Decision:**

Accept (regular)

**Comment:**

This paper introduces an informed asymmetric actor-critic (IAAC) framework. The key idea is that the critic can use arbitrary state-dependent privileged signals during training, rather than the full state. The authors prove that policy gradients remain unbiased even under these more general assumptions. The approach is evaluated on 6 benchmark navigation tasks.

This paper is well written and studies a topic that has a potential to be followed. The reviewers gave the paper scores 4x Weak Accept. The rebuttal addressed two main concerns:

* **Evaluation on more complex problems:** The authors reported additional results on more complex problems from POPGym benchmark. They also promised to report wall-clock times.

* **Motivation:** Simply put, why should anybody care about training a critic on less historic data when the prevalent approach in machine learning is training on long histories? The answer is increased computational and statistical efficiency.

All reviewers suggested weak acceptance of this paper and so do I.